# Influenza virus uses mGluR2 as an endocytic receptor to enter cells

Zixin Ni[1,2], Jinliang Wang [1,2], Xiaofei Yu[1], Yifan Wang[1], Jingfei Wang[1], Xijun He[1], Chengjun Li [1], Guohua Deng[1], Jianzhong Shi[1], Huihui Kong[1], Yongping Jiang[1], Pucheng Chen[1], Xianying Zeng[1], Guobin Tian[1], Hualan Chen [1] ✉ & Zhigao Bu [1] ✉

Influenza virus infection is initiated by the attachment of the viral haemagglutinin (HA) protein to sialic acid receptors on the host cell surface. Most virus particles enter cells through clathrin-mediated endocytosis (CME). However, it is unclear how viral binding signals are transmitted through the plasma membrane triggering CME. Here we found that metabotropic glutamate receptor subtype 2 (mGluR2) and potassium calcium-activated channel subfamily M alpha 1 (KCa1.1) are involved in the initiation and completion of CME of influenza virus using an siRNA screen approach. Influenza virus HA directly interacted with mGluR2 and used it as an endocytic receptor to initiate CME. mGluR2 interacted and activated KCa1.1, leading to polymerization of F-actin, maturation of clathrin-coated pits and completion of the CME of influenza virus. Importantly, mGluR2-knockout mice were significantly more resistant to different influenza subtypes than the wild type. Therefore, blocking HA and mGluR2 interaction could be a promising host-directed antiviral strategy.

Influenza A viruses are categorized into different subtypes based on the different antigenicity of their two surface glycoproteins, haemagglutinin (HA) and neuraminidase (NA). H1N1, H2N2 and H3N2 influenza viruses have caused four pandemics with over 50 million fatalities[1,2]. H5 and H7 avian influenza viruses have caused over 2,600 human cases with over 1,000 deaths globally[3], and numerous disease outbreaks in birds, with the loss of at least 422 million domestic poultry since 2005[4].

Influenza virus infection is initiated by the viral HA binding to a sialic acid receptor that presents as a glycoprotein or glycolipid on the host cell surface. A previous study suggested that glycolipids facilitate attachment but are not involved in internalization, and influenza virus specifically requires N-linked glycoprotein for entry[5]. Other studies reported that the binding of influenza virus to sialylated cell surface proteins does not always result in virus internalization[6,7], suggesting that only certain glycoproteins mediate viral internalization.

Influenza virus internalizes into host cells through clathrin-mediated endocytosis (CME)[8–10], clathrin-independent endocytosis[11,12] and macropinocytosis[13]. Most influenza virus particles enter cells via CME[8–10], which is a receptor-mediated, signal-regulated process that transports various transmembrane proteins (that is, receptors) and their extracellular ligands from the plasma membrane of eukaryotic cells into the cytoplasm[14]. Although many host factors involved in influenza virus infection have been identified[8,15–22], it remains unclear which transmembrane protein is used by influenza virus to transduce the signal of viral binding across the plasma membrane to initiate CME. In this study, we found two membrane proteins––potassium calcium-activated channel subfamily M alpha 1 (KCa1.1) encoded by *KCNMA1* and metabotropic glutamate receptor 2 (mGluR2) encoded by *GRM2*––that are crucial for initiating and completing the CME of influenza virus. Our findings open a new door for the development of host-directed strategies against influenza virus.

## Results

### Identification of K⁺ channel KCa1.1

Ion channels play key roles during virus infection[23]. The L-type calcium channel was reported to affect influenza virus entry, although it is not

[1]State Key Laboratory for Animal Disease Control, Harbin Veterinary Research Institute, Chinese Academy of Agricultural Sciences, Harbin, People's Republic of China. [2]These authors contributed equally: Zixin Ni, Jinliang Wang. ✉e-mail: chenhualan@caas.cn; buzhigao@caas.cn

clear which step of viral entry the channel affects[17]. To screen for more ion channels involved in influenza virus infection, we performed an RNA interference (RNAi) assay in A549 cells with 1,014 short interfering RNAs (siRNAs) targeting 338 human genes. The cells were transfected with corresponding siRNA, and 72 h later, they were infected with A/WSN/1933 (H1N1)[21] (hereafter referred to as H1N1 virus) at a multiplicity of infection (MOI) of 0.01. Non-targeting scramble siRNA (siControl) and siRNA targeting H1N1 nucleoprotein (siNP) were used as negative and positive controls, respectively. The 50% tissue culture infectious doses (TCID$_{50}$) of viruses in the supernatants were titrated at 24 h post-infection (p.i.) in Madin-Daby canine kidney (MDCK) cells (Extended Data Fig. 1a). A gene was selected as a positive hit if (1) the viral titre was reduced at least tenfold compared with siControl-transfected cells, (2) the $P$ value was lower than 0.05 and (3) at least two of three siRNAs fulfilled criteria (1) and (2). Using these criteria, 37 genes (Supplementary Table 1)—16 genes related to potassium channels (Fig. 1a,b), 12 genes related to calcium channels or calcium-regulated unselected cation channels, 3 genes encoded ATPases, 1 gene related to sodium channels and 5 genes predicted to have ion channel activity[24] (Extended Data Fig. 1b)—were identified as positive hits, and 6 of them, *KCNE2*, *CACNA1C*, *GRIN3A*, *CHRNA1*, *ATP6V1B2* and *KCNMA1*, were previously identified by others[25,26]. These results suggest that different ion channels are involved in influenza virus infection.

In this study, we focused on the roles of potassium channels in influenza virus replication. Among the 16 identified genes related to potassium channels, we found that the downregulation of KCa1.1 had the greatest effect on the replication of H1N1 virus in A549 cells (Fig. 1b–d). To further confirm this observation, we performed drug inhibition assays with paxilline, a functional antagonist of KCa1.1 (ref. 27), in A549 cells. We found that paxilline treatment had no effect on cell viability (Fig. 1e), but significantly inhibited the replication of H1N1 virus in a dose-dependent manner (Fig. 1f). These results show that KCa1.1 plays an important role in influenza virus infection.

## KCa1.1 regulates influenza virus entry

To investigate which stage of influenza virus replication is affected by KCa1.1, we performed a time-of-addition assay in A549 cells. The relative viral RNA (vRNA) levels in cell lysates were quantified by quantitative PCR (qPCR) at 6 h p.i. We found that paxilline treatment performed at −1 h or 0 h p.i. significantly reduced vRNA levels in cells compared with control cells, but treatment performed at 1 h, 2 h or 3 h p.i. showed no difference in vRNA levels compared with control cells (Fig. 1g), indicating that KCa1.1 plays a key role in the early steps of influenza virus infection.

The early steps of influenza virus infection involve attaching to sialic acid receptors at the cell surface and internalization into the cell. To investigate whether KCa1.1 affects influenza virus attachment in cells with different treatments, the vRNA levels of attached viruses were quantified by qPCR. We found that NA treatment significantly reduced virus binding, whereas KCa1.1 silencing or paxilline treatment did not (Fig. 1h), indicating that KCa1.1 has no effect on influenza virus binding.

To investigate whether KCa1.1 is involved in the internalization step, the vRNA levels of internalized viruses in cells with different treatments were quantified by qPCR. The vRNA levels inside the paxilline-treated cells or KCa1.1-silenced cells were significantly lower than those of control cells (Fig. 1i), indicating that activated KCa1.1 is important for the influenza virus internalization, which was confirmed by a microscopy-based assay with z-stacks[18] that quantitatively detects antibody-labelled viruses that are retained on cell membranes after an internalization period under unpermeabilized conditions (Fig. 1j,k). However, the vRNA level inside KCa1.1-overexpressing cells was comparable to that of control cells (Fig. 1l). To test whether the HA of influenza virus interacts with KCa1.1, we constructed a recombinant H1N1 HA with a six-histidine tag (HA–His)[17] (Extended Data Fig. 2a,b). The HA–His purified from eukaryotic cells was mixed with KCa1.1–Myc-transfected

HEK293 cell lysates for a pull-down assay. We found that KCa1.1 did not interact with HA–His (Extended Data Fig. 2c), implying that other host factors may be cooperating with KCa1.1 during this process.

## Cooperation between mGluR2 and KCa1.1

A previous study identified a series of host factors that interact with KCa1.1 (ref. 28); we found that seven of these host factors—mGluR2, CPNE9, ADCY5, NTSR2, GABRA5, SCN8A and MPDZ—are cellular transmembrane proteins expressed in alveolar epithelial cells and may activate KCa1.1 (ref. 29). To evaluate their effect on influenza virus replication, two specific siRNAs were designed for each of these seven candidates for RNAi assays. We found that knockdown of four genes—*mGluR2*, *CPNE9*, *ADCY5* and *NTSR2*—led to a decrease in viral titres in A549 cells, whereas knockdown of the other three genes had no effect on influenza virus replication (Fig. 2a). We next evaluated the effect of these four genes on influenza virus binding and internalization. We found that none of these host factors affected influenza virus binding, and only mGluR2 affected influenza virus internalization (Fig. 2b), which was further confirmed with the microscopy-based assay (Fig. 2c).

To investigate whether KCa1.1 and mGluR2 work through an interdependent process or independently, we compared the H1N1 virus replication in paxilline-treated A549 cells, mGluR2-silenced A549 cells and paxilline-treated mGluR2-silenced A549 cells. We found that simultaneous blocking of KCa1.1 and mGluR2 had no additive effect on the reduction of viral titres (Fig. 2d), ruling out the possibility of independent operation.

To investigate whether KCa1.1 and mGluR2 cooperate to facilitate the internalization of influenza virus, we validated their interaction by performing a co-immunoprecipitation assay in plasmid-transfected HEK293 cells (Fig. 2e), and then compared the vRNA levels of H1N1 virus attached and internalized in A549 cells, which endogenously express KCa1.1 and mGluR2, that were transfected with the mGluR2-expression plasmid alone or co-transfected with the KCa1.1-expression plasmid. We found that overexpression of mGluR2 alone did not increase vRNA levels of attached and internalized viruses, whereas overexpression of both KCa1.1 and mGluR2 did not increase vRNA levels of attached viruses, but significantly increased vRNA levels of internalized viruses (Fig. 2f). These results suggest that KCa1.1 and mGluR2 jointly promote the internalization of influenza viruses.

## mGluR2 internalizes with influenza virus

To investigate whether KCa1.1 and mGluR2 are internalized with influenza virus, we quantified the abundance of KCa1.1 and mGluR2 on the cellular membrane before and after H1N1 virus internalization using a microscopy-based assay. The fluorescence intensity of KCa1.1 on the membrane of H1N1-virus-internalized cells was comparable to that of the uninfected control A549 cells or H1N1-virus-bound cells (Fig. 2g), but the fluorescence intensity of mGluR2 on the membrane of H1N1-virus-internalized cells was significantly lower than that of the uninfected control A549 cells or H1N1-virus-bound cells (Fig. 2h), indicating that mGluR2, but not KCa1.1, internalizes with influenza virus. Of note, in the KCa1.1-silenced cells, the fluorescence intensity of mGluR2 on the membrane of H1N1-virus-internalized cells was comparable to that of the control A549 cells and the virus-bound cells (Fig. 2i), indicating that KCa1.1 is required for the internalization of mGluR2.

## mGluR2 serves as an endocytic receptor

In addition to CME, influenza virus also exploits caveolin-mediated endocytosis[11,12] and macropinocytosis[13] to enter cells. Clathrin heavy chain (CLTC), caveolin 1 (CAV1) and Rac family small GTPase 1 (RAC1) are key molecules involved in CME, caveolin-mediated endocytosis and macropinocytosis[30], respectively. To investigate which pathway in influenza virus internalization involves mGluR2, we first performed microscopy-based assays to evaluate the abundance of mGluR2 on the membrane of CLTC-, CAV1- and RAC1-silenced A549 cells as well

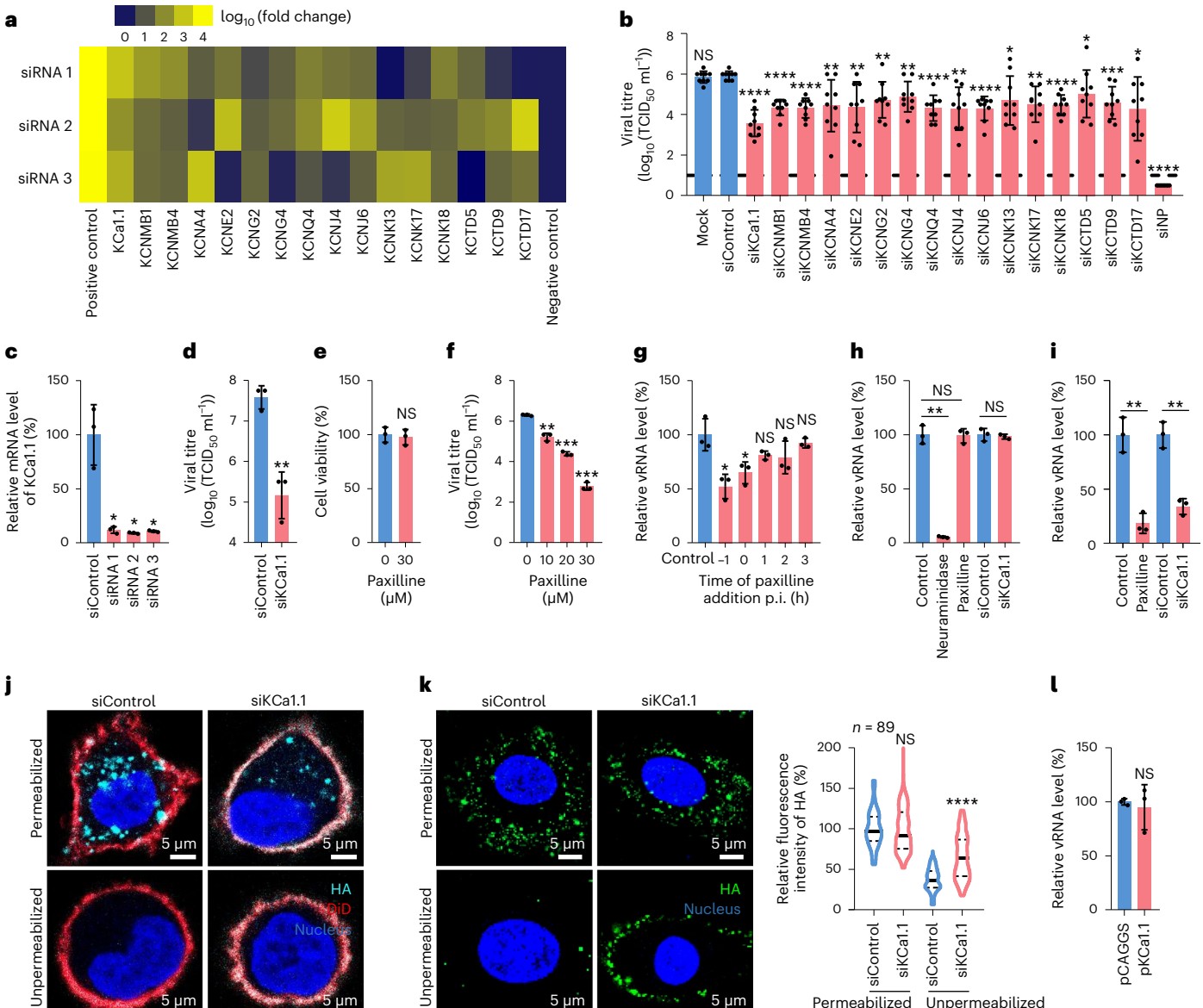

**Fig. 1 | KCa1.1 plays an important role in influenza virus internalization.**
**a**, Potassium channels involved in influenza virus replication were identified by screening siRNA libraries of human ion channels. **b**, Viral titres in different potassium channel-knockdown A549 cells. The data were summarized from the screen by combining the results from different siRNAs presented in **a. c**, The mRNA level of KCa1.1 in KCa1.1-knockdown A549 cells was significantly lower than that in control cells. **d**, KCa1.1 knockdown significantly reduced H1N1 virus replication in A549 cells. **e**, Paxilline treatment did not affect the viability of A549 cells. **f**, Paxilline treatment significantly reduced H1N1 virus replication in A549 cells. **g**, Paxilline affected the early stage of influenza infection. **h**, KCa1.1-knockdown and paxilline treatment did not affect the attachment of H1N1 virus to A549 cells. **i**, KCa1.1-knockdown or paxilline treatment affected the internalization of H1N1 virus into A549 cells. **j**, KCa1.1-knockdown affected the internalization of H1N1 virus confirmed by a three-dimensional microscopy-based assay under permeabilized and unpermeabilized conditions with z-stacks and a membrane marker. Cell nucleus, blue; viral HA, turquoise; DiD, red. **k**, KCa1.1 knockdown affected the internalization of H1N1 virus into A549 cells confirmed by a microscopy-based assay under permeabilized and unpermeabilized conditions. Cell nucleus, blue; viral HA, green. The solid line represents median and dashed lines represent quartiles of the data in the violin graphs of **k. l**, KCa1.1 overexpression did not increase H1N1 virus internalization. The images in **j** and **k** are representative of three independent experiments. Error bar in panels **b**–**i** and **l** indicates the standard deviation. The data shown in **a**–**i**, **k** and **l** are means ± s.d. ($n = 3$ biologically independent experiments). Statistical analysis was performed using unpaired, two-tailed Student's $t$-test. NS, not significant. $^*P < 0.05$; $^{**}P < 0.01$; $^{***}P < 0.001$; $^{****}P < 0.0001$. Exact $P$ values are available in Source Data.

as control A549 cells after H1N1 virus internalized. The fluorescence intensity of mGluR2 on the membrane of CLTC-silenced cells was significantly higher than that on the control cells, whereas the fluorescence intensity of mGluR2 on the membrane of CAV1- or RAC1-silenced cells was comparable to that on the control cells (Fig. 3a), indicating that mGluR2 is involved in the CME of influenza virus.

mGluR2, a seven-transmembrane N-linked glycoprotein of the G-protein-coupled receptor (GPCR) superfamily[31], is widely expressed in human tissues (BioGPS database)[32] (Extended Data Fig. 3). mGluR2 is

activated by extracellular ligands and transduces intracellular signals via interactions with G proteins[33–35]. As mGluR2 does not affect influenza virus binding but is required for influenza virus internalization, we speculated that mGluR2 may serve as an endocytic receptor and transduce the signal of viral binding across the plasma membrane to initiate CME. To test this hypothesis, we first tested the interaction of HA and mGluR2, and found that the HA–His was successfully pulled down by mGluR2 (Fig. 3b). This was further confirmed using stimulated emission depletion (STED) super-resolution microscopy, which showed that mGluR2 and H1N1 virus

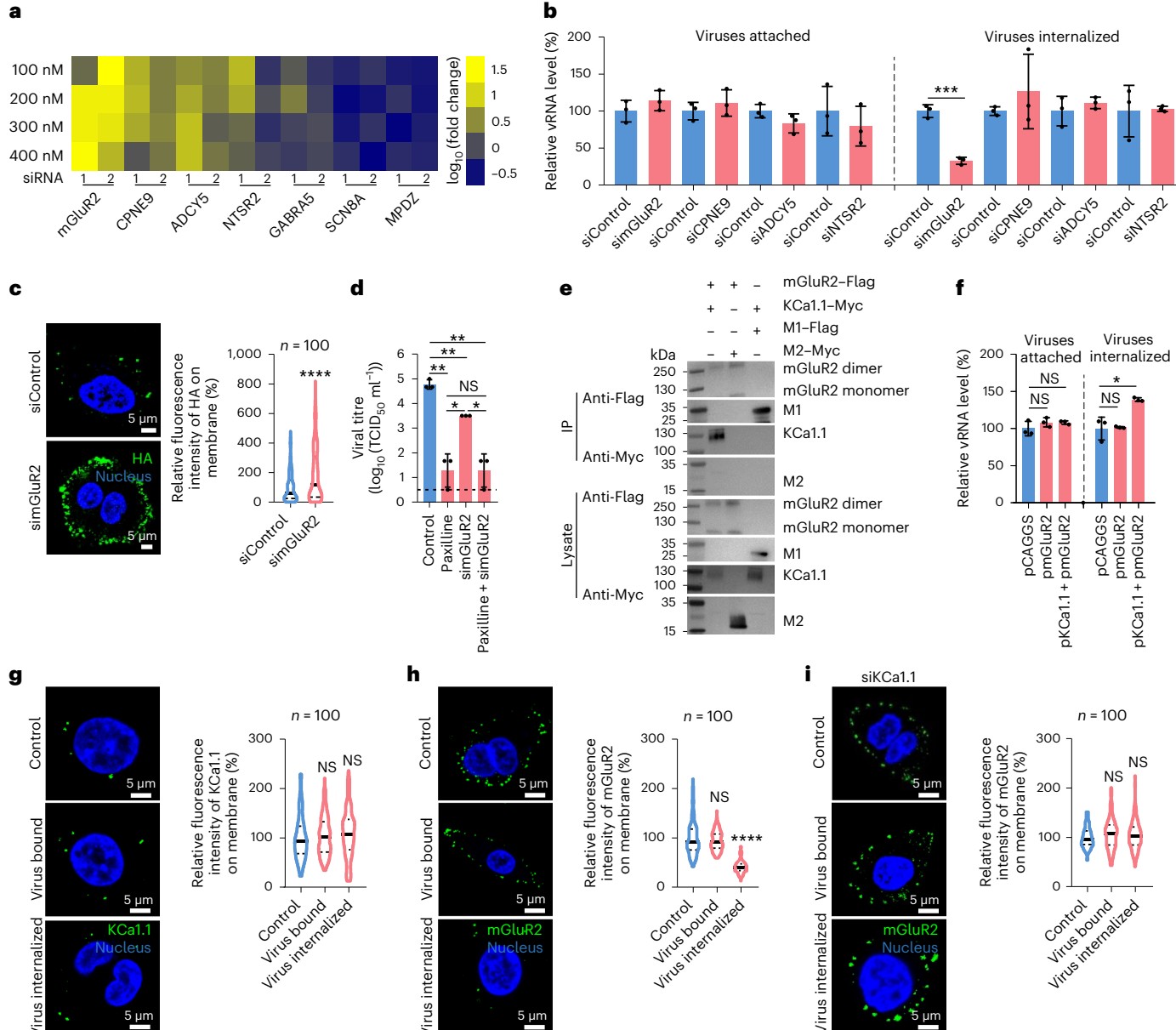

**Fig. 2 | mGluR2 plays an important role in influenza virus internalization.**
**a**, Effect of seven different membrane proteins that interact with KCa1.1 on influenza virus replication evaluated using an RNAi assay. **b**, mGluR2 knockdown reduced the internalization of H1N1 virus into A549 cells. **c**, mGluR2 knockdown reduced the internalization of H1N1 virus into A549 cells, as confirmed using a microscopy-based assay. Cell nucleus, blue; viral HA, green. **d**, H1N1 virus replication in A549 cells with different treatments. **e**, Interaction of mGluR2 and KCa1.1 confirmed by co-immunoprecipitation with agarose beads coupled with anti-Flag antibodies. **f**, Co-overexpression of KCa1.1 and mGluR2 increased H1N1 virus internalization. **g**, KCa1.1 did not internalize with H1N1 virus into A549 cells. Cell nucleus, blue; KCa1.1, green. **h**, mGluR2 internalized with H1N1

virus into A549 cells. Cell nucleus, blue; mGluR2, green. **i**, KCa1.1 knockdown prevented internalization of mGluR2 with influenza virus into A549 cells. Cell nucleus, blue; mGluR2, green. The microscopy-based assays shown in **c**, **g**, **h** and **i** were performed under unpermeabilized conditions. The solid line represents median and dashed lines represent quartiles of the data in the violin graphs of **c** and **g**–**i**. The images in **c**, **e**, **g**, **h** and **i** are representative of three independent experiments. Error bar in panels **b**, **d** and **f** indicates the standard deviation. The data shown in **a**–**d** and **f**–**i** are means ± s.d. (n = 3 biologically independent experiments). Statistical analysis was performed using the unpaired, two-tailed Student's t-test; NS, not significant. *P < 0.05; **P < 0.01; ***P < 0.001; ****P < 0.0001. Exact P values are available in the Source Data.

particles co-localized in A549 cells during viral internalization (Fig. 3c). Moreover, by use of immunoelectron microscopy, we detected mGluR2 on the cell membrane of different stages of influenza virus endocytosis, including the cell membrane of influenza-virus-containing invagination pits and the cell membrane of matured endocytic vesicles, which are consistent with those of the clathrin-coated pits (CCPs) and clathrin-coated vehicles (CCVs)[36] (Fig. 3d).

To investigate whether the interaction between mGluR2 and HA is important for influenza virus internalization, we performed

an infectivity neutralization assay with the purified ectodomain of mGluR2 with a glutathione S-transferase (GST) tag (mGluR2–GST). The mGluR2–GST did not affect the cell viability at the highest concentration used (Fig. 3e); it inhibited H1N1 virus infection in a dose-dependent manner in A549 cells (Fig. 3f). We further performed an antibody blocking assay with a monoclonal antibody against the ectodomain of mGluR2 (mGluR2–ma), and found that mGluR2–ma inhibited H1N1 virus infection in a dose-dependent manner (Fig. 3g,h). These results show that mGluR2 directly interacts with the HA protein and serves as

an endocytic receptor for the CME of influenza virus. mGluR2 did not affect the endocytosis of adenovirus type 5 or transferrin (Extended Data Fig. 4a and Fig. 3f,h,i), although they also enter cells by CME[37,38], suggesting that mGluR2-mediated CME is cargo specific.

## KCa1.1 regulates F-actin polymerization

KCa1.1 neither interacts with the HA of influenza virus nor internalizes with the virus, suggesting that KCa1.1 may regulate other factors involved in the CME of influenza virus. KCa1.1 channels could be activated by $Ca^{2+}$, membrane potential or membrane stretch[39–41]. Stretch sensitivity of smooth muscle KCa1.1 channels involves interactions with F-actin[39,41], and F-actin dynamics are essential for efficient completion of CCPs[42]. Therefore, we speculated that changes in $Ca^{2+}$ or membrane stretching induced by influenza virus binding may activate KCa1.1 and thus regulate F-actin polymerization. To test this concept, we evaluated F-actin polymerization in A549 control cells; A549 cells treated with paxilline or cytochalasin D (cyto D), an inhibitor of actin polymerization[43]; and KCa1.1-knockdown A549 cells, following H1N1 virus internalization. The F-actin dye phalloidin-488 staining revealed reduced fluorescence intensity of phalloidin in the KCa1.1-knockdown cells, paxilline-treated cells and cyto-D-treated cells compared with A549 control cells (Fig. 4a). Moreover, like knockdown of KCa1.1, cyto D treatment also abolished the internalization of mGluR2 (Fig. 4b) and thereby prevented influenza virus internalization (Fig. 4c). These results show that the activation of KCa1.1 positively affects F-actin polymerization and thereby facilitates the CME of influenza virus.

To further show the role of KCa1.1 and mGluR2 in the CME of influenza virus, we observed the formation of CCPs of influenza virus at 1 min post-binding (p.b.) or 6 min p.b. in A549 cells, KCa1.1-knockdown A549 cells, mGluR2-knockdown A549 cells and A549 cells treated with chlorpromazine (a CME inhibitor) using transmission electron microscopy. We found that 51.5%, 17.5%, 20.2% and 2.5% of the viral particles that attached to the membrane of the control cells, KCa1.1-knockdown cells, mGluR2-knockdown cells and chlorpromazine-treated cells, respectively, formed clear CCPs at 1 min p.b. (Fig. 4d), and 91.7%, 33.8%, 44.4% and 8.0% of the viral particles that attached to the membrane of the control cells, KCa1.1-knockdown cells, mGluR2-knockdown cells and chlorpromazine-treated cells, respectively, formed clear CCPs at 6 min p.b. (Fig. 4d). Notably, KCa1.1, but not mGluR2, knockdown also reduced CCP formation of adenovirus type 5 (Extended Data Fig. 4b). These results show that KCa1.1 and mGluR2 play important roles in the CME of influenza virus. Taken together, our findings allow us to propose a model of the roles and collaboration of KCa1.1 and mGluR2 in the CME of influenza virus (Fig. 4e).

## Role of mGluR2 for H5 and H7 viruses

Our studies showed that H1N1 virus uses mGluR2 as an endocytic receptor for its CME and that this mGluR2-mediated CME is regulated by KCa1.1. It is unclear whether viruses of other subtypes also internalize cells through this mechanism. We therefore evaluated the role of KCa1.1 and mGluR2 on H5 and H7 virus internalization using the viruses A/duck/Guangdong/S1330/2016 (H5N6) (ref. 44) and A/chicken/

Guangdong/SD008/2017 (H7N9)-PB2/627K (ref. 45) (hereafter referred to as H5N6 virus and H7N9 virus, respectively). We found that knockdown of KCa1.1 or mGluR2 significantly suppressed the infection and internalization of both viruses (Extended Data Fig. 5a–d and Extended Data Fig. 6a–d). We showed that mGluR2 interacts with the HA gene of H5N6 and H7N9 viruses, and that the soluble protein mGluR2–GST or mGluR2–ma inhibited infection of both viruses (Extended Data Fig. 6e–j). Together, these results suggest that CME with mGluR2 as an endocytic receptor and regulated by KCa1.1 may be a common mechanism for influenza virus internalization.

## Studies in mGluR2-knockout mice

Our in vitro studies show that mGluR2 plays a direct role in the CME of influenza virus. To investigate whether mGluR2 affects influenza virus infection in vivo, we evaluated the replication and lethality of different influenza viruses in wild-type and mGluR2-knockout (*mGluR2[−/−]*) mice. Groups of 16 wild-type C57BL/6J mice and *mGluR2[−/−]* mice were inoculated intranasally with 10 50% mouse lethal dose ($MLD_{50}$) of the indicated influenza virus. Three mice in each group were euthanized on days 3 and 5 p.i., and their organs, including the nasal turbinates, lungs, brain, kidneys and spleens, were collected for virus titration. The remaining 10 mice in each group were monitored for body weight changes and survival for 2 weeks.

H1N1 virus was detected in the nasal turbinates, lungs and kidneys of all wild-type mice on days 3 and 5 p.i., and in the spleen of one mouse on day 5 p.i. (Fig. 5a). However, the virus was detected only in the nasal turbinates and lungs of *mGluR2[−/−]* mice, with titres being notably or significantly lower than those in the wild-type mice (Fig. 5a). The wild-type mice lost nearly 29% of their body weight and all died on day 9 p.i. (Fig. 5b,c). The *mGluR2[−/−]* mice had a similar weight loss, but 60% of them survived (Fig. 5b,c).

H5N6 virus was detected in all five test organs of wild-type mice and in four test organs (not the brain) of *mGluR2[−/−]* mice, and the titres in the organs of *mGluR2[−/−]* mice were significantly lower than those of wild-type mice (Fig. 5d). The wild-type mice lost nearly 23% of their body weight, and all died within 12 days p.i. (Fig. 5e,f). However, the *mGluR2[−/−]* mice lost about 12% body weight and had a survival rate of 70% (Fig. 5e,f).

H7N9 virus was detected in four of the five tested organs (not the spleen) of wild-type mice and in only the nasal turbinates and lungs of *mGluR2[−/−]* mice. The titres in the *mGluR2[−/−]* mice were significantly lower than those of wild-type mice (Fig. 5g). The wild-type mice lost nearly 19% of their body weight, and all died within 11 days p.i., whereas the *mGluR2[−/−]* mice lost about 8% of their body weight and had a survival rate of 70% (Fig. 5h,i). These studies show that mGluR2 knockout significantly weakened the replication of influenza viruses in the nasal turbinates and lungs of mice, and prevented the replication of H5 and H7 viruses in the brain of mice.

Immunohistological study and dual staining analysis indicated that viral-antigen-positive cells (MUC1[+] type II alveolar epithelial cells, AQP5[+] type I epithelial cells and S100A9[+] macrophages) in the lungs of *mGluR2[−/−]* mice were notably less than those in the wild-type mice

---

**Fig. 3 | Influenza virus uses mGluR2 as an endocytic receptor for its CME. a**, Knockdown of CLTC significantly prevented mGluR2 internalization in H1N1-virus-internalized A549 cells. Cell nucleus, blue; mGluR2, green. **b**, Direct interaction of mGluR2 and HA shown using a pull-down assay with the agarose beads coupled with anti-Flag antibodies. **c**, Co-localization of mGluR2 and HA confirmed using a STED super-resolution microscopy assay. Cell nucleus, blue; mGluR2, red; and viral HA, green. **d**, mGluR2 molecules detected on the cell membrane where influenza virus attached, on the membrane of different stages of CCPs of influenza virus and on the membrane of matured CCVs of influenza virus by means of immunoelectron microscopy in mGluR2-overexpressing A549 cells. Green arrow, mGluR2; red arrow, influenza virus. Scale bar, 100 nm. **e,g**, mGluR2–GST (**e**) or mGluR2–ma (**g**) did not affect the viability of A549 cells.

**f,h**, mGluR2–GST (**f**) or mGluR2–ma (**h**) treatment significantly reduced H1N1 virus replication but did not reduce adenovirus type 5 replication in A549 cells. **i**, Internalization of transferrin was affected by chlorpromazine (a CME inhibitor) but not by mGluR2-GST or mGluR2-ma. The microscopy-based assay shown in **a** and **i** was performed under unpermeabilized conditions and that shown in **c** was performed under permeabilized conditions. The solid line represents median and dashed lines represent quartiles of the data in the violin graphs of **a** and **i**. The images in **a–d** and **i** are representative of three independent experiments. Error bar in panels **e–h** indicates the standard deviation. The data shown in **a** and **c–i** are means ± s.d. (*n* = 3 biologically independent experiments). Statistical analysis was performed using the unpaired, two-tailed Student's *t*-test. NS, not significant. *$P < 0.05$; ***$P < 0.001$; ****$P < 0.0001$. Exact $P$ values are available in Source Data.

(Fig. 6 and Extended Data Fig. 7), suggesting that mGluR2 knockout reduces the overall replication level of influenza virus, but does not affect viral cell tropism.

## Discussion

In this study, we identified two proteins, KCa1.1 and mGluR2, important for the CME of influenza virus. mGluR2 interacts with HA and serves

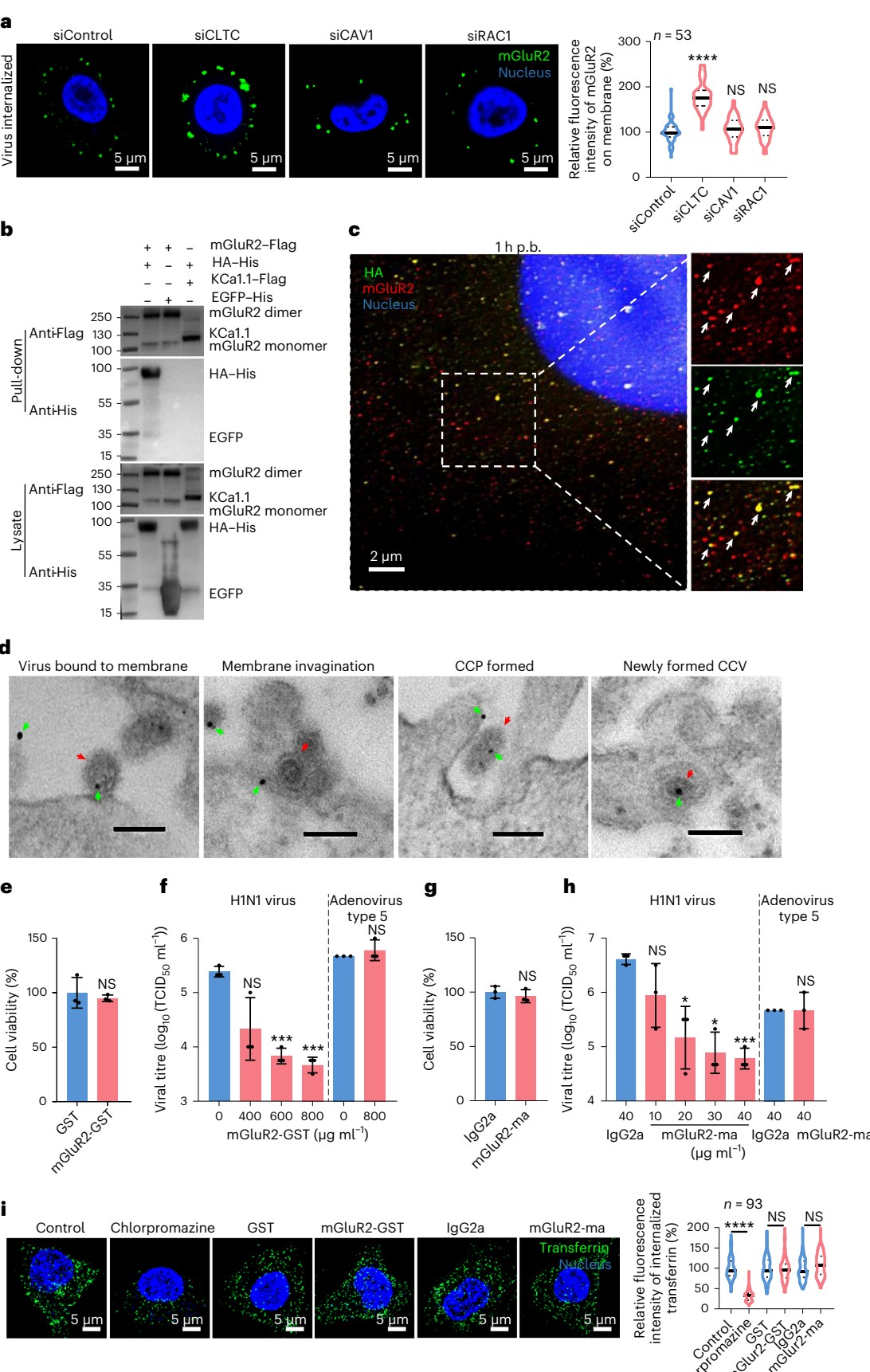

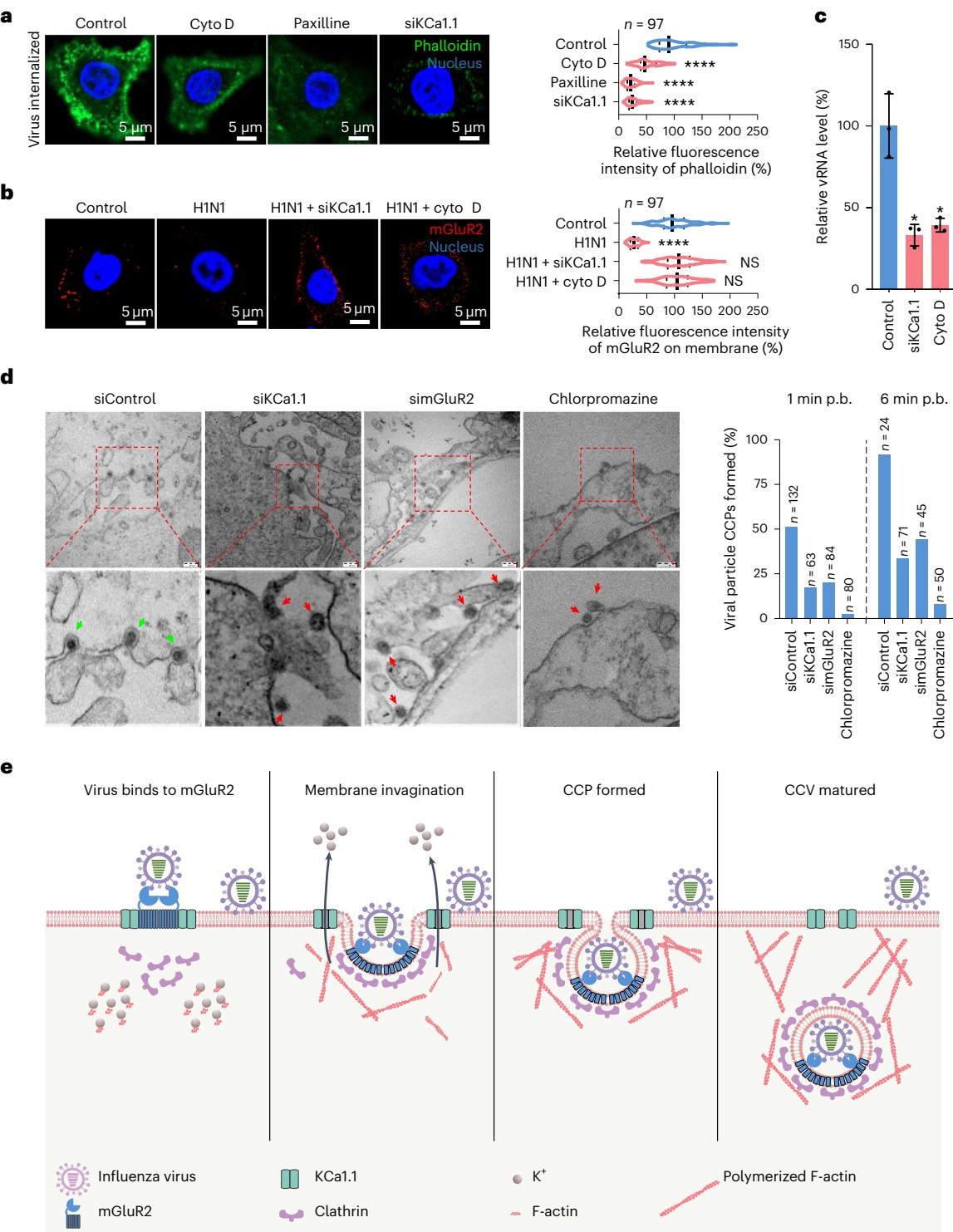

**Fig. 4 | KCa1.1 facilitates influenza virus internalization by regulating F-actin polymerization. a**, KCa1.1-knockdown, cyto D or paxilline treatment significantly reduced F-actin polymerization in H1N1-virus-internalized A549 cells. Cell nucleus, blue; phalloidin, green. **b**, Cyto D treatment or KCa1.1 knockdown prevented the influenza-virus-associated internalization of mGluR2 into A549 cells. Cell nucleus, blue; mGluR2, red. **c**, Cyto D or KCa1.1-knockdown treatment significantly reduced the internalization of H1N1 virus into A549 cells. **d**, KCa1.1 or mGluR2 knockdown and chlorpromazine treatment reduced the formation of CCPs of influenza virus in A549 cells. Green arrow, influenza virus CCP; red arrows, influenza virus attached. Scale bar, 200 nm. **e**, Model of the roles of mGluR2 and

KCa1.1 in the CME of influenza virus. The microscopy-based assays shown in **a** and **b** were performed under unpermeabilized conditions. The solid line represents median and dashed lines represent quartiles of the data in the violin graphs of **a** and **b**. The images in **a**, **b** and **d** are representative of three independent experiments. Error bar in panels **c** indicates the standard deviation. The data shown in **a**–**c** are means ± s.d. ($n$ = 3 biologically independent experiments). Statistical analysis was performed using the unpaired, two-tailed Student's $t$-test. NS, not significant. *$P$ < 0.05; ****$P$ < 0.0001. Exact $P$ values are available in Source Data.

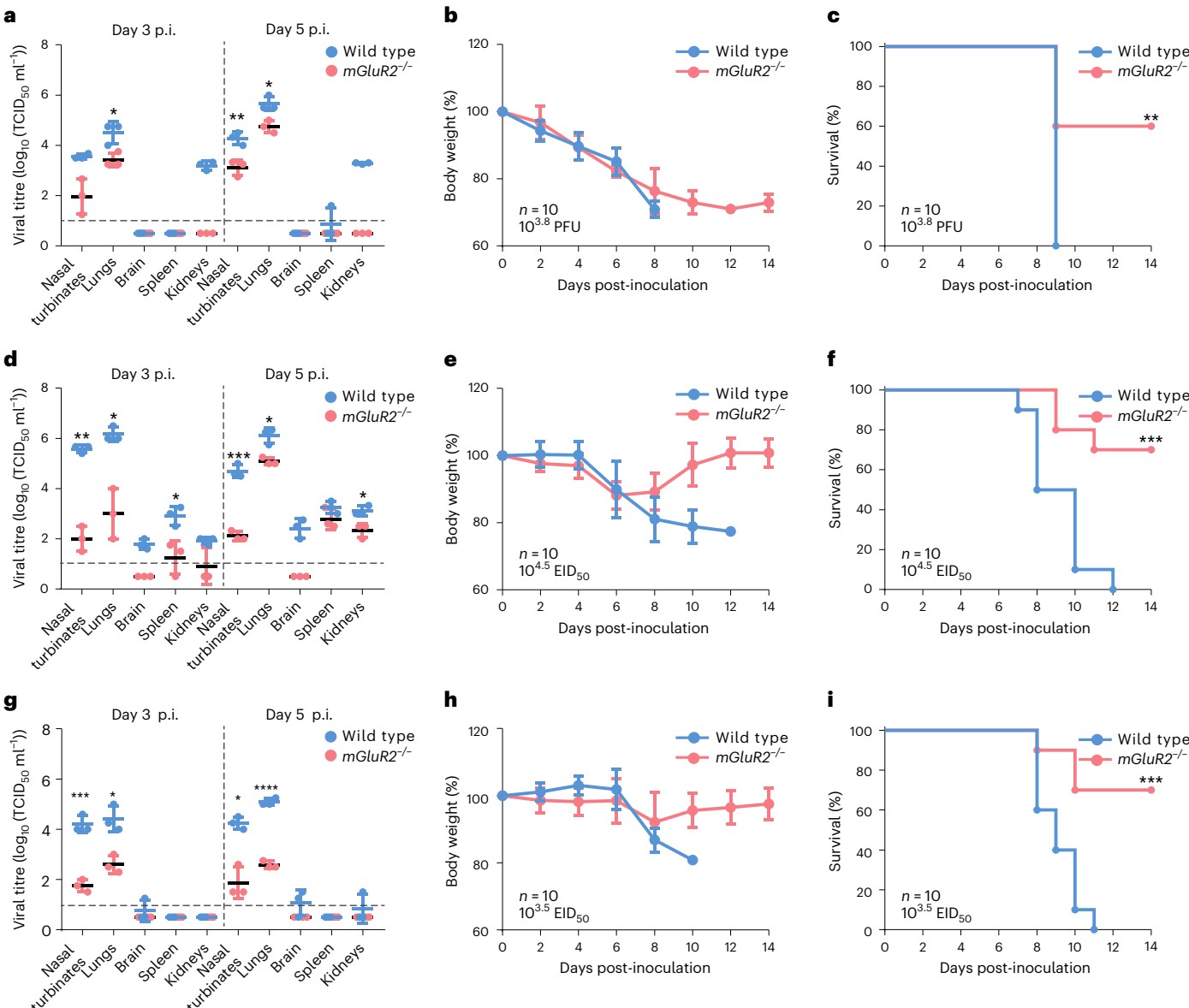

**Fig. 5 | Replication and virulence of different influenza viruses in wild-type C57BL/6J and *mGluR2⁻/⁻* mice. a–i,** Groups of 16 wild-type C57BL/6J mice and *mGluR2⁻/⁻* mice were intranasally inoculated with 10 MLD$_{50}$ of H1N1 virus (**a,b,c**), H5N6 virus (**d,e,f**) or H7N9 virus (**g,h,i**). **a,d,g,** Three mice in each group were euthanized on days 3 and 5 p.i., respectively, and their organs indicated were collected for virus titration in cells. **b,c,e,f,h,i,** The remaining 10 mice in each group were evaluated for body weight change (**b,e,h**) and survival (**c,f,i**) for

2 weeks. Error bar in panels **a, b, d, e, g** and **h** indicates the standard deviation. Data in **a, d** and **g** are means ± s.d. Statistical analysis was performed using the unpaired, two-tailed Student's *t*-test. The log-rank (Mantel–Cox) test was used to analyse the statistical difference in survival rates between the wild-type and *mGluR2⁻/⁻* mice. *$P < 0.05$; **$P < 0.01$; ***$P < 0.001$; ****$P < 0.0001$. PFU, plaque-forming unit; EID$_{50}$, 50% egg infectious dose. Exact $P$ values are available in Source Data.

as an endocytic receptor for the CME of influenza virus, and KCa1.1 facilitates this process by regulating the polymerization of F-actin, which is required for the maturation of CCPs of influenza virus and completion of the CME of influenza virus. Moreover, we found that *mGluR2* knockout dramatically increases the resistance of mice to different subtypes of influenza viruses. Although influenza virus can enter cells through different ways, and other signalling pathways may also mediate the CME of influenza virus[26], our study suggests that KCa1.1- and mGluR2-mediated CME is an important pathway for influenza virus internalization into cells.

N-linked glycoprotein mGluR2 facilitates influenza virus CME, which is consistent with a previous report that influenza virus requires N-linked glycoprotein for entry[5]. However, it was not clear whether the endocytosis was dependent on the sialic acid of N-linked

glycoprotein. There are five potential glycosylation sites (PGSs) in the ectodomain of mGluR2, and we proved that some of these sites are glycosylated because mGluR2 in which these PGSs were eliminated (mGluR2-mutant) moved faster than the wild type on western blotting (Extended Data Fig. 8a–c). Although we do not know whether the glycoprotein mGluR2 carries sialic acid, the elimination of PGSs does not affect the interaction of mGluR2 with influenza virus HA (Extended Data Fig. 8b,c), suggesting that their interaction could be independent of the sialic acid of mGluR2. Glycosylation is not a prerequisite for mGluR2 to initiate influenza virus endocytosis, but it is crucial for stabilizing and increasing cellular mGluR2 levels, as overexpressing of mGluR2-mutant did not completely restore influenza virus infection in mGluR2-knockdown cells owing to lower expression levels (Extended Data Fig. 8d–f). Our study suggests that influenza virus binding and

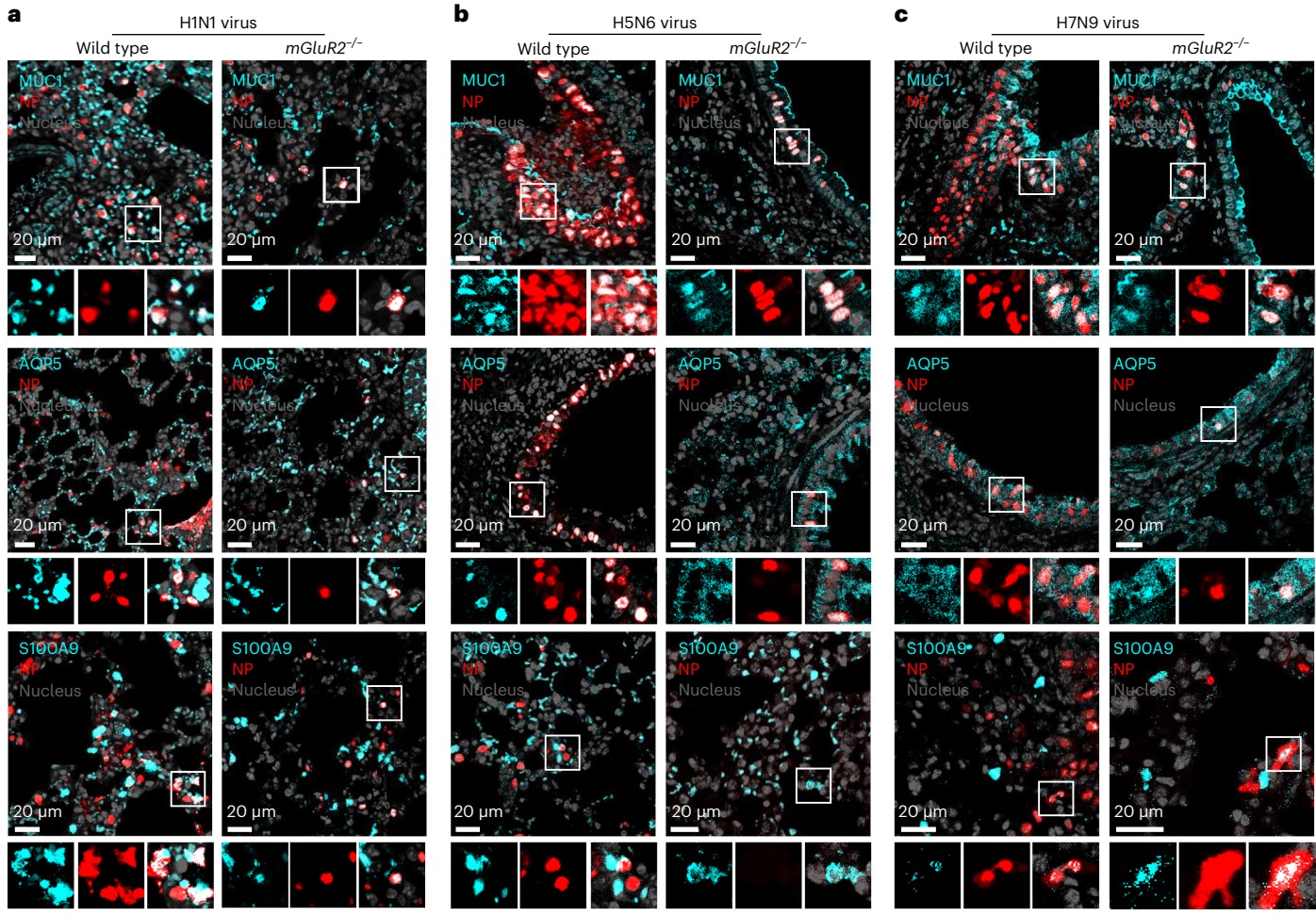

**Fig. 6 | Identification of cell types in the lungs of wild-type and *mGluR2*⁻/⁻ mice infected by influenza virus. a–c,** Dual staining was performed on the lung sections of mice infected with H1N1 virus (**a**), H5N6 virus (**b**) or H7N9 virus (**c**) to identify the type II alveolar epithelial cells (MUC1-positive cells), type I alveolar epithelial cells (AQP5-positive cells) and macrophages (S100A9-positive cells) that were viral antigen positive. Cell nucleus, grey; viral NP, red; MUC1/AQP5/S100A9, turquoise. The images in **a–c** are representative of three independent experiments.

endocytosis are mediated by different receptors, and that glycosylation is not a prerequisite for the N-linked glycoprotein to exert its endocytic receptor function.

Several host factors have been reported to influence the influenza virus internalization[8,15–20,22], but it is unclear which internalization pathway they are involved in except for FFAR2, which interacts with certain GPCR kinases and thereby influences the CME of influenza virus[21]. mGluR2 and KCa1.1 are both required for the CME of influenza virus. Due to the importance of KCa1.1 in cell biology, we were unable to generate KCa1.1-knockout mice and test its role in influenza virus replication in vivo. The *mGluR2*⁻/⁻ mice can reproduce healthily and stably, suggesting that lack of mGluR2 may not affect the growth and development of animals. The pathogenicity of influenza viruses to humans is directly related to their extensive replication in the lungs[46–48], and intracerebral replication is the main cause of mortality in avian species infected with highly pathogenic avian influenza viruses[49,50]. Our study showed that mGluR2 knockout significantly reduces influenza virus replication in mouse lungs and prevents influenza virus replication in the mouse brain. Therefore, blocking the interaction between influenza virus HA and mGluR2 would be a promising host-directed antiviral strategy.

## Methods

### Ethics statement

Animal studies were carried out in strict accordance with the recommendations in the Guide for the Care and Use of Laboratory Animals of the Ministry of Science and Technology of the People's Republic of China. The protocols were approved by the Committee on the Ethics of Animal Experiments of Harbin Veterinary Research Institute (HVRI) of the Chinese Academy of Agricultural Sciences (CAAS; approval number IACUC-2022-220719-01-GJ).

### Biosafety statements and facility

All experiments with live H5N6 and H7N9 viruses were conducted in the enhanced animal biosafety level 3 (ABSL3+) facility in HVRI of CAAS, and the experiments with live H1N1 virus were conducted in the enhanced ABSL2+ facility in HVRI of CAAS, which are approved for such use by the Ministry of Agriculture of China and the China National Accreditation Service for Conformity Assessment. All mouse experiments were approved by the Review Board of HVRI, CAAS.

### Mice, cells and viruses

C57BL/6J (*Mus musculus*) *mGluR2* knockout (*mGluR2*⁻/⁻) mice were generated as described previously[51]. Sex- and age-matched wild-type C57BL/6J mice were purchased from Vital River Laboratories. The mice were housed in ventilated cages (with a maximum of eight mice per cage) with food and water ad libitum. Mice were housed under 12 h light and dark cycles in a controlled environment maintained at 24 °C and 40–60% humidity.

HEK293 cells were obtained from American Type Culture Collection (ATCC, CRL-1573) and maintained in Dulbecco's modified

Eagle's medium containing 10% fetal bovine serum (FBS). A549 cells were obtained from ATCC (CCL-185) and maintained in Ham's F-12K (Kaighn's) Medium with 10% FBS. MDCK cells were obtained from ATCC (PTA-6500) as were 293E cells (CRL-1573); both cell lines were maintained in Dulbecco's modified Eagle's medium containing 5% FBS. All culture media contained penicillin–streptomycin (10,000 U ml⁻¹). The cells were cultured at 37 °C with 5% $CO_2$.

A/WSN/1933 (H1N1) (H1N1 virus)[21], A/duck/Guangdong/S1330/2016 (H5N6) (H5N6 virus)[44], A/chicken/Guangdong/SD008/2017 (H7N9)-PB2/627K (ref. 45) (H7N9 virus) and adenovirus type 5 were maintained in our laboratory. Influenza viruses were propagated in 10-day-old embryonated chicken eggs, and adenovirus type 5 was propagated in 293E cells.

### Antibodies, reagents and chemicals

Chicken anti-H5-HA polyclonal antibody (pAb), chicken anti-H7–HA pAb (1:500) and mouse anti-nucleoprotein monoclonal antibody (mAb; 1:200) were generated in our laboratory using conventional methods. Rabbit anti-H1–HA pAb antibody (catalogue number (CN) 11692-T62, 1:300) was from Sino Biological. Rabbit anti-KCa1.1 pAb (CN APC-151, 1:100) was from Alomone Labs. Mouse anti-mGluR2 mAb (CN sc271654, 1:200) was from Santa Cruz Biotechnology. Rabbit anti-V5-tag mAb (CN 13202S, 1:1,000) was from Cell Signaling. Rabbit anti-Flag-tag pAb (CN A00170, 1:1,000) and rabbit anti-Myc-tag pAb (CN A00172, 1:1,000) were from Genscript. Rabbit anti-6-His pAb (CN SAB4301134, 1:1,000), goat anti-chicken IgY coupled with horseradish peroxidase (HRP) (CN A9046, 1:2,000) and goat anti-rabbit IgG coupled with gold (10 nm; CN G7402, 1:1,000) were from Sigma-Aldrich. Mouse IgG2a mAb (CN 0103-01) was from Southern Biotech. Rabbit anti-Na⁺/K⁺ ATPase mAb (CN ab76020, 1:500), goat anti-mouse IgG coupled with Alexa Fluor 568 (CN ab175473, 1:300), rabbit anti-S100A9 mAb antibody (CN ab242945, 1:800) and goat anti-mouse IgG coupled with Alexa Fluor 488 (CN ab150113, 1:300) were from Abcam. Goat anti-rabbit IgG coupled with Alexa Fluor 488 (CN A11034, 1:300) and goat anti-rabbit IgG coupled with HRP (CN 31460, 1:2,000) were from Invitrogen. Goat anti-mouse Abberior STAR RED (CN 20831PK-3, 1:200) and goat anti-rabbit Abberior STAR ORANGE (CN 20831PK-6, 1:200) were from Abberior. Rabbit anti-MUC1 pAb antibody (CN bs-1497R, 1:100) was from Bioss. Rabbit anti-AQP5 pAb antibody (CN A03085, 1:100) was from Boster. Goat anti-mouse IgG coupled with Cy3 (CN A0521, 1:100) was from Beyotime. Horse anti-mouse IgG coupled with HRP (CN MP-7802-15, 1:5) and goat anti-rabbit IgG coupled with fluorescein (CN FI-1000, 1:200) were from VectorLabs.

The opti-MEM (CN 31985070), ion channel siRNA library (CN 4397915), Lipofectamine RNAiMAX Transfection Reagent (CN 13778150), 4′,6-diamidino-2-phenylindole (DAPI; CN D3571), Pierce IP Lysis Buffer (CN 87788), Halt Protease Inhibitor Cocktail (CN 87786), anti-Myc-tag antibody-conjugated agarose beads (CN 20168), live cell imaging solution (CN A14291DJ), and Alexa Fluor 488-labelled transferrin (CN T13342) were from Invitrogen. CellTiter-Glo kit (CN G9242) was from Promega. L-1-tosylamide-2-phenylmethyl chloromethyl ketone-treated trypsin (CN T1426) and anti-Flag-tag antibody-conjugated agarose beads (CN A2220) were from Sigma-Aldrich. ExFect Transfection Reagent (CN T101-02), ChamQ Universal SYBR qPCR Master Mix (CN Q711-03), QuickChange Site-Directed Mutagenesis kit (CN C215-01) and ExFect Transfection Reagent (CN T101-02) were from Vazyme. PVDF membrane (CN ISEQ00010) and immobilon crescendo western HRP substrate (CN WBLUR0500) were from Merck-Millipore. SeaPlaque agarose (CN 50100) was from Lonza. RNA extraction kit (CN: DP419) was from TIANGEN. First Strand cDNA Synthesis Kit (CN FSK-101) was from TOYOBO. Paxilline (CN B6920) was from Apexbio. Chlorpromazine (CN HY-12708), cyto D (CN HY-N6682) was from MedChemExpress. Minute Plasma Membrane Protein Isolation and Cell Fractionation Kit (CN SM-005) was from Invent Biotechnologies. Benzonase (CN C2001) was from HaiGene. The 4–12%

ExpressPlus PAGE Gel (CN M41210C) was from Genscript. Protein G agarose (CN 11243233001) and neuraminidase (CN 52163721) were from Roche. Phalloidin iFlour 488 (CN ab176753) was from Abcam. Cell Plasma Membrane Staining Kit with DiD (CN C1995S) was from Beyotime.

### Plasmids

The KCa1.1 overexpressing plasmid pKCa1.1 was constructed by inserting the cDNA of KCa1.1 into the mammalian expression vector pCAGGS with a Myc or Flag tag at the C-terminus. The overexpressing plasmid pmGluR2 was constructed by inserting the cDNA of mGluR2 into pCAGGS with a Flag tag or a Myc tag at the C-terminus. The pmGluR2 mutant was constructed by introducing mutations to eliminate the five PGSs at positions 203, 286, 338, 402 and 547 using the QuickChange Site-Directed Mutagenesis kit. The HA genes of H5N6 virus and H7N9 virus were respectively cloned into pCAGGS. The HA gene of H1N1 virus was cloned into pCAGGS with a V5 tag at the C-terminus. The M1 gene of H1N1 virus was cloned into pCAGGS with a Flag tag at the C-terminus. The M2 gene of H1N1 virus was cloned into pCAGGS with a Myc tag at the C-terminus.

### Cell viability assay

Cell viability was determined using the CellTiter-Glo kit following the manufacturer's instructions. In brief, A549 cells treated with chemicals, antibodies or soluble proteins with the indicated concentration were maintained in opaque 96-well plates for the indicated time. Then, the medium was replaced with 100 µl of CellTiter-Glo reagent per well, and the cells were incubated for 10 min at room temperature. The luminescence was then measured using a GloMax 96 Microplate Luminometer (Promega).

### qPCR

To quantify the level of mRNA of KCa1.1 or the vRNA in cells after different treatments, the total RNA of cells was extracted using a total RNA extraction kit. The cDNA was generated with a First Strand cDNA Synthesis Kit. The qPCR was generated with ChamQ Universal SYBR qPCR Master Mix, with the specific primers listed in Supplementary Table 2. The qPCR was conducted using the Applied Biosystems QuantStudio 5 Real-Time PCR System (Thermo Fisher). The $2^{-\Delta\Delta Ct}$ method was used to calculate the relative gene level with GAPDH as the internal control.

### RNAi assay

The RNAi assay was performed in 96- or 6-well plates. For the 96-well plates, 40 µl of opti-MEM containing 0.4 µl of Lipofectamine RNAiMAX Transfection Reagent was added to the wells in which the siRNAs were arrayed. After 30 min of incubation at room temperature, A549 cells (8,000 cells per well) in a volume of 60 µl were added. For the 6-well plates, 500 µl of opti-MEM containing 4.5 µl of Lipofectamine RNAiMAX Transfection Reagent was added to the wells in which the siRNAs were arrayed. After 30 min of incubation at room temperature, A549 cells ($2 \times 10^5$ cells per well) in a volume of 1.5 ml were added. The siRNA-transfected A549 cells were maintained at 37 °C for 72 h before they were infected with influenza virus.

The ion channel RNAi screening assay was performed in 96-well plates as described above. The siRNA-transfected A549 cells were maintained at 37 °C for 72 h before they were infected with H1N1 virus (MOI 0.01). The supernatants of the samples were collected for virus titration in MDCK cells at 24 h p.i. Wild-type A549 cells, non-targeting siControl-transfected cells and the siNP of H1N1-transfected cells were used as mock, negative and positive controls, respectively. The viral titres were normalized with those from siControl-transfected A549 cells.

### Drug inhibition assays

For the paxilline inhibition assay, wild-type or siRNA-transfected A549 cells were pre-treated with drugs at different concentrations for 1 h

before they were infected with influenza virus (MOI 5). At 24 h p.i., the supernatants were collected for virus titration in MDCK cells.

For the paxilline-addition assay, A549 cells seeded in 6-well plates were treated with paxilline (25 μM) at −1, 0, 1, 2 or 3 h p.i. (MOI 5) and then cultured for 6 h at 37 °C. The cells were then lysed to measure the vRNA level by qPCR.

### Viral binding assay

For the viral binding assay, the drug-treated and siRNA-transfected A549 cells in 6-well plates were infected with influenza virus (MOI 50) at 4 °C for 1 h. The cells were then washed three times with cold PBS and lysed to measure the vRNA levels by qPCR.

### Viral internalization assay

For the viral internalization assay, the drug-treated and siRNA-transfected A549 cells in 6-well plates were infected with influenza virus (MOI 50) and incubated at 4 °C for 1 h. The cells were washed 3 times with cold PBS and maintained at 37 °C for 2 h. Then, the cells were washed 8 times with cold acidic PBS (pH 1.3) to remove the virus retained on the surface of the plasma membrane. After that, the cells were lysed for the qPCR assay to analyse the vRNA in the cells. For the microscopy-based assay, virus-bound A549 cells were washed 3 times with cold PBS and maintained at 37 °C for 1 h. The cells were fixed and the cell surface HA was analysed using the confocal laser scanning microscopy assay with the indicated antibodies.

### Microscopy-based assays

For the three-dimensional confocal laser scanning microscopy assays, control A549 cells and KCa1.1-knockdown cells infected with H1N1 virus (MOI 50) were incubated at 4 °C for 1 h. After being washed 3 times with cold PBS, the cells were then incubated at 37 °C for 1 h. Then, the cells were fixed with 4% polyformaldehyde (PFA) for 30 min and the cellular membrane was stained with DiD at 37 °C for 15 min. The cells were washed and permeabilized with 0.1% saponion (this step was skipped for the unpermeabilized conditions). The permeabilized or unpermeabilized cells were blocked with 5% BSA in PBS for 1 h and incubated with specific primary antibody for 2 h. The cells were washed three times with PBS and incubated with the indicated fluorophore-coupled secondary antibody for 1 h. After three washes with PBS, the cells were incubated with DAPI for 15 min to stain the cell nucleus. Images were acquired using a confocal laser scanning microscope 980 (Carl Zeiss) and scanned with z-stacks of 40 layers.

For the confocal laser scanning microscopy assays, A549 cells with different treatments were fixed with 4% PFA for 30 min and then treated with 0.5% Triton X-100 for 30 min (the latter step was skipped for the unpermeabilized conditions). The permeabilized or unpermeabilized cells were then processed as described above except for staining the membrane with DiD. The fluorescence intensity of each cell was calculated using the ZEN software (Carl Zeiss).

### Western blotting

The plasmid-transfected cells were washed twice with cold PBS and lysed with Pierce IP Lysis Buffer containing Halt Protease Inhibitor Cocktail and Benzonase for 1 h at 4 °C. Cell lysates were centrifuged at 13,523 $g$ at 4 °C for 10 min. The supernatants were then mixed with protein sample loading buffer and boiled for 10 min. The samples were loaded onto a 4–12% ExpressPlus PAGE Gel and separated by electrophoresis. Proteins were transferred to a PVDF membrane. The PVDF membrane was blocked with 5% skim milk in PBS containing 0.1% Tween-20 (PBST buffer) and incubated with the indicated primary antibody at room temperature for 1 h. The membrane was then washed three times with PBS and incubated with the indicated HRP secondary antibody. After the membrane was washed three times with PBST buffer, the target protein bands were visualized with Immobilon Crescendo Western HRP substrate using the Odyssey infrared imaging

system (Li-Cor BioSciences). The intensities of bands were analysed using Image J software (version 1.53t).

### Co-immunoprecipitation

For the co-immunoprecipitation assay, the cells with different treatments were lysed as described for the western blotting assay and the supernatants of the cell lysates were mixed with 40 μl of protein G Agarose for 4 h at 4 °C on a flip shaker. The protein G beads were removed by centrifugation, and the supernatants were collected and mixed with anti-Flag-tag antibody-conjugated agarose beads or anti-Myc-tag antibody-conjugated agarose beads for 12 h at 4 °C on the flip shaker. After conjugation, the beads were washed five times with cold NP-40. The beads were re-suspended in PBS, mixed with protein sample loading buffer and boiled for 10 min, then subjected to SDS-PAGE and assessed by western blot analysis.

### Pull-down assay

For the pull-down assay, the recombinant HA–His protein expressed in eukaryotic cells was purified using FriendBio Technology. Plasmid overexpressing (Flag-tag) HEK293 cells were lysed, and the supernatants of the cell lysates were incubated with anti-Flag antibody-conjugated agarose beads at 4 °C for 2 h. The beads were then washed with cold NP-40 and incubated with HA–His protein overnight. After conjugation, the beads were washed ten times with cold NP-40. The beads were re-suspended in PBS and mixed with protein sample loading buffer and boiled for 10 min, then subjected to SDS-PAGE and assessed by western blot analysis.

### STED super-resolution microscopy

We performed STED microscopy to observe the co-localization of HA and mGluR2 during viral internalization. Virus-internalized A549 cells were fixed and permeabilized as described above. The permeabilized cells were then incubated with primary antibodies and secondary antibodies, respectively. STED images were taken using an Abberior Instruments Stedycon equipped with an inverted IX83 microscope and a ×100/1.45 oil objective (Olympus). All acquisition operations were controlled using the Stedycon software (Abberior Instruments).

### Immunoelectron microscopy localization of mGluR2 and influenza virus

The mGluR2-Myc-overexpressing A549 cells were infected with H1N1 virus (MOI 200) and incubated at 4 °C for 1 h; the bound virus was allowed to internalize at room temperature for 5 min after binding. The cells were then fixed with 1% glutaraldehyde and 4% PFA, and incubated at 4 °C for 2 h. The samples were dehydrated with 50%, 70%, 90% and 100% DMF, respectively, at 4 °C for 15 min; then, the samples were processed for embedding with UV irradiation polymerization at −20 °C for 10 days. Then, the samples were incubated with the primary antibody of Myc-tag for 40 min at room temperature. After abundant washes with distilled water, the samples were incubated with the secondary antibody of goat anti-rabbit IgG coupled with gold (10 nm) for 40 min at room temperature. After abundant washes with distilled water, the samples were stained with uranyl acetate for 10 min, and images of the localization of mGluR2 and influenza virus were acquired by means of transmission electron microscopy.

### Antibody blocking assay

A monoclonal antibody against mGluR2 (mGluR2-ma, 200 mg ml⁻¹; Santa Cruz Biotechnology, CN sc271654) was used for the antibody blocking assay. A549 cells were incubated with 0.12 ml of opti-MEM containing different concentrations of mGluR2−ma (10 μg ml⁻¹, 20 μg ml⁻¹, 30 μg ml⁻¹, 40 μg ml⁻¹) or control isotype antibody IgG2a (40 μg ml⁻¹) at 4 °C for 1 h, and then were washed three times with opti-MEM containing the corresponding antibodies. The cells were then infected with the antibody–virus mixture and incubated at 4 °C for 1 h. After another

round of three washes, the cells were incubated with 0.1 ml of opti-MEM containing the corresponding antibodies at 37 °C. At 48 h p.i., the supernatants of the samples were collected for virus titration in cells.

### Infectivity neutralization assay

The infectivity neutralization assay was performed on A549 cells with the soluble protein of mGluR2 (mGluR2–GST) that was expressed and purified using FriendBio Technology[52]. Viruses were mixed with different concentrations of mGluR2–GST (400 µg ml$^{-1}$, 600 µg ml$^{-1}$ or 800 µg ml$^{-1}$) or control GST protein (800 µg ml$^{-1}$) in 0.1 ml of opti-MEM at 4 °C for 1 h. A549 cells in 96-well plates were infected with the protein–virus mixture and incubated at 37 °C for 1 h. The cells were washed three times with opti-MEM to remove the unbound viruses and were then maintained with 0.1 ml of opti-MEM medium containing the corresponding proteins at 37 °C for 24 h. The supernatants of the samples were collected for virus titration in cells.

### Transferrin internalization assay

A549 cells with different treatments were washed twice with PBS and maintained in opti-MEM for 2 h at 37 °C. Then, the cells were incubated with 5 µg ml$^{-1}$ Alexa Fluor 488-labelled transferrin for 30 min. To stop the internalization, the cells were incubated with cold wash buffer (50 mM glycine, 100 mM NaCl, pH 3.0) for 2 min. Then, the cells were fixed with 4% PFA for 30 min and the unpermeabilized cells were incubated with DAPI for 15 min to stain the cell nucleus. Images were acquired using a confocal laser scanning microscope 980 (Carl Zeiss).

### F-actin polymerization assay

The viral internalization assay was performed in drug-treated or siRNA-transfected A549 cells; the cells were fixed and washed three times with PBS and incubated with Phalloidin iFlour 488 for 1 h. Then, the F-actin polymerization in the cells was analysed by performing a confocal laser scanning microscopy assay.

### Detection of CCPs during virus internalization using transmission electron microscopy

The process of influenza virus internalization was observed using transmission electron microscopy. Briefly, A549 cells treated with KCa1.1 knockdown, mGluR2 knockdown or chlorpromazine (30 µM) were infected with viruses (MOI 200) and incubated at 4 °C for 1 h. The bound virus was allowed to internalize at room temperature for the indicated times; then, the cells were fixed with 2.5% glutaraldehyde for 16 h. Ultrathin sections were prepared on carbon-coated 100-mesh copper grids and observed in a Hitachi-7650 transmission electron microscope at an operating voltage of 80 kV. The formation of viral particle CCPs was calculated.

### Mouse experiment

Groups of 16 wild-type and *mGluR2*$^{-/-}$ mice were intranasally infected with 10 MLD$_{50}$ of H1N1 virus, H5N6 virus or H7N9 virus. Three mice in each group were euthanized on days 3 and 5 p.i., and their organs were collected for virus titration in MDCK cells. The remaining 10 mice in each group were monitored for body weight changes and survival for 14 days.

Groups of two wild-type and *mGluR2*$^{-/-}$ mice were intranasally infected with 10 MLD$_{50}$ of H1N1 virus, H5N6 virus or H7N9 virus. The mice were euthanized on day 3 p.i., and their lungs were collected for immunohistochemistry study and cell type identification as described by others[47]. For immunohistochemistry study, mouse anti-nucleoprotein mAb antibody and horse anti-mouse IgG coupled with HRP were used as primary antibody and secondary antibody, respectively. For cell type identification, the lung sections of mice were first incubated with the mouse anti-nucleoprotein mAb antibody as the primary antibody and visualized using goat anti-mouse IgG coupled with Cy3. The sections were then washed with TBS and incubated with either the rabbit anti-AQP5 mAb antibody, rabbit anti-MUC1

mAb antibody or rabbit anti-S100A9 mAb antibody. The cell markers (AQP5, MUC1 and S100A9) were visualized using goat anti-rabbit IgG coupled with fluorescein. Sections were mounted with Vectashield (Vector Laboratories) and observed using the Zeiss LSM700 confocal laser scanning microscope.

### Infection restoration assay

A549 cells or mGluR2-knockdown A549 cells were transfected with pCAGGS, pmGluR2 or pmGluR2 mutant for 24 h, respectively. Then, the cells were infected with H1N1 virus (MOI 0.01) and incubated at 37 °C for 24 h. The supernatants of the samples were collected for virus titration in cells.

### Membrane protein extraction

A549 cells were transfected with pCAGGS, pmGluR2 or pmGluR2 mutant for 24 h. Membrane proteins were isolated from $3 \times 10^8$ cells using the Minute Plasma Membrane Protein Isolation and Cell Fractionation kit and following the manufacturer's instructions. The plasma membrane proteins were analysed by western blotting.

### Flow cytometry

A549 cells were transfected with pmGluR2 or pmGluR2 mutant for 24 h and collected in a 1.5 ml tube. Then, the cells were washed three times with FACS wash buffer (PBS containing 2% FCS) and fixed with 4% PFA at room temperature for 15 min. The cells were then permeabilized with 0.5% Triton X-100 for 30 min (this step was skipped for the unpermeabilized conditions). The permeabilized or unpermeabilized cells were incubated with mouse anti-mGluR2 mAb (1:100) as the primary antibody and goat anti-mouse IgG coupled with Alexa Fluor 488 (1:1,000) as the secondary antibody. The cells were analysed using a FC-500 flow cytometer (Beckman Coulter). The fluorescence intensity of mGluR2 was analysed using FlowJo software.

### Statistics and reproducibility

For all the bar and violin graphs, data are shown as the mean ± s.d. All statistical analyses were performed using GraphPad Prism software v.8.00 for the unpaired, two-tailed Student's *t*-test as indicated in the figure legends. The log-rank (Mantel–Cox) method was used for survival analysis. Differences in means were considered statistically significant at $P < 0.05$, and significance levels are as follows: *$P < 0.05$; **$P < 0.01$; ***$P < 0.001$; and ****$P < 0.0001$; NS, not significant. Data distribution was assumed to be normal, but this was not formally tested. No statistical methods were used to predetermine sample sizes, but our sample sizes are similar to those reported in previous publications[17,20,21,51,52]. Data collection was not performed blind to the conditions of the experiments, but analysis was blinded. No animals or data points were excluded from the analyses.

### Reporting summary

Further information on research design is available in the Nature Portfolio Reporting Summary linked to this article.

## Data availability

The authors declare that the data supporting the findings of this study are available within the Article and Supplementary Tables. The mRNA sequencing data of mGluR2 (Extended Data Fig. 3) in human tissues were obtained from BioGPS database (http://biogps.org/#goto=genereport&id=2912). Source data are provided with this paper.

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

## Acknowledgements

This work was supported by the National Key R&D Program of China (2021YFD1800200 to H.C., 2021YFC2301700 to J.S., 2023YFC2605001 to Jingfei W.), the National Natural Science Foundation of China (grant number 32202773 to Jinliang W., 32192451 to J.S.), the Central Public-Interest Scientific Institution Basal Research Fund (number Y2023XK10 to Jinliang W.), the Innovation Program of Chinese Academy of Agricultural Sciences (CAAS-CSLPDCP-202301 to J.S., CAAS-CSLPDCP-202401 to J.W.) and the earmarked fund for CARS-41 (CARS-41 to G.T.).

## Author contributions

Z.N., J.W., X.Y., Y.W., J.W., X.H., C.L., G.D., J.S., H.K., Y.J., P.C., X.Z. and G.T. performed the experiments; Z.N., J.W., H.C. and Z.B. analysed the data; Z.N., J.W., H.C. and Z.B. designed the study; H.C. and Z.B. wrote the paper.

## Competing interests

The authors declare no competing interests.

## Additional information

**Extended data** is available for this paper at https://doi.org/10.1038/s41564-024-01713-x.

**Correspondence and requests for materials** should be addressed to Hualan Chen or Zhigao Bu.

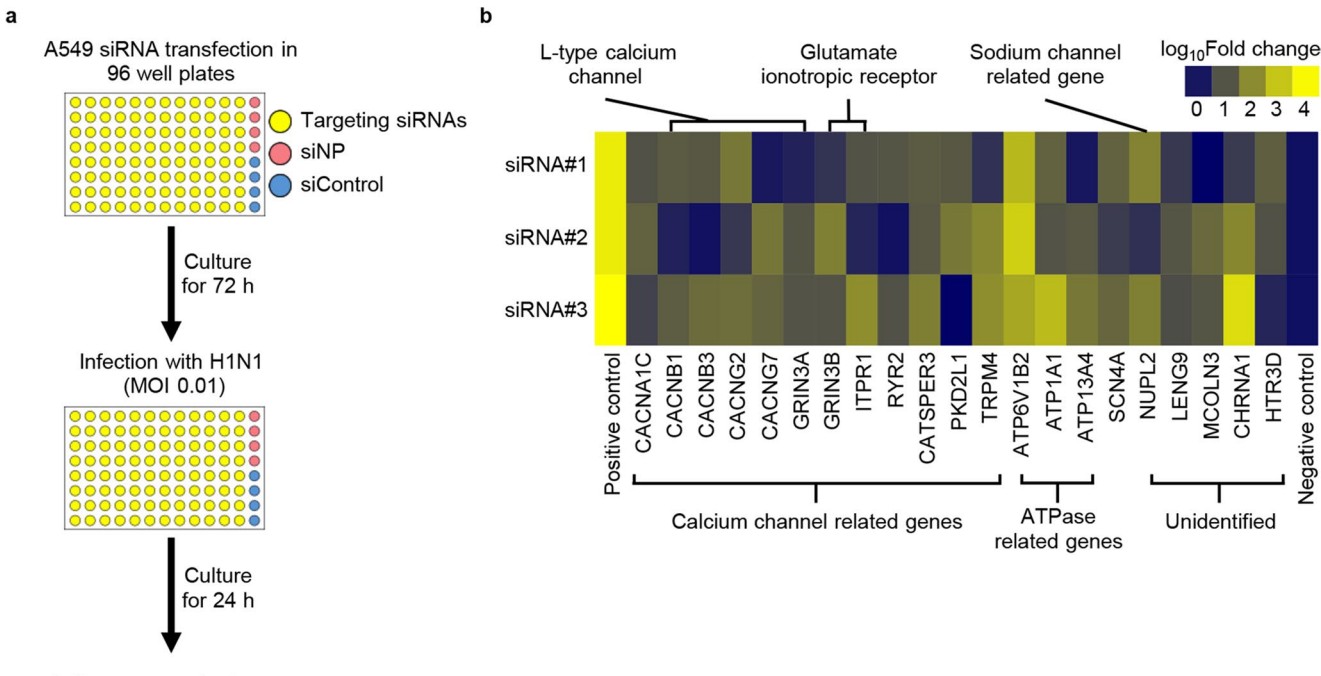

**Extended Data Fig. 1 | Result of the ion channel RNAi screen. a**, Outline of the ion channel RNAi screening procedures. Wild-type A549 cells, non-targeting scramble siRNA (siControl)-transfected cells, and siRNA targeting the nucleoprotein (siNP) of H1N1-transfected cells were used as mock, negative, and positive controls, respectively. **b**, 21 ion channels are involved in the replication of H1N1 virus in A549 cells from the human ion channel siRNA library screen.

Indicated siRNAs were individually transfected into A549 cells for 72 h and followed by infection with H1N1 virus at an MOI of 0.01 in three independent experiments. Infectious viral particles in the supernatants were determined at 24 p.i. as $TCID_{50}$/mL and analyzed by calculating the fold-change of inhibition. The data are means ± s.d. (n = 3 biologically independent experiments).

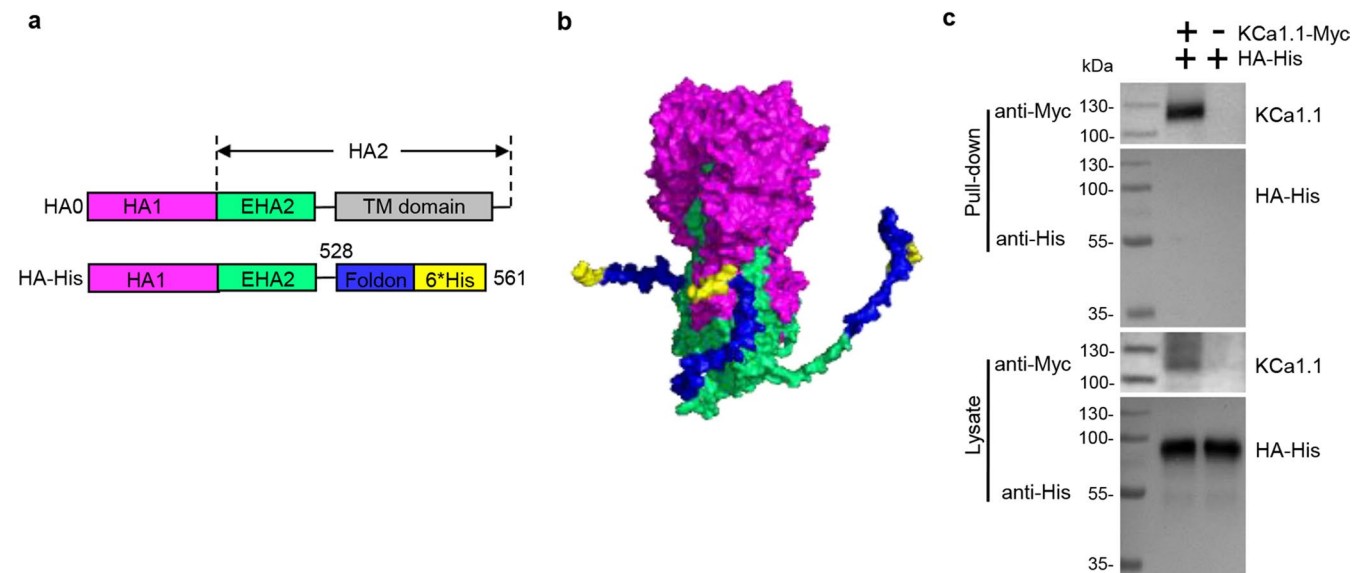

**Extended Data Fig. 2 | KCa1.1 does not interact with the purified HA of H1N1 virus by use of a pull-down assay. a**, Schematic representation of the domain structures for wild-type HA of H1N1 and a 6×His-tagged form of HA (HA-His). **b**, Structural prediction of HA-His by Alphafold2 and visualized by Pymol. **c**, KCa1.1 does not interact with HA. Direct interaction of KCa1.1 and HA was tested by use of a pull-down assay with the anti-Myc antibody coupled agarose beads. The images in panel **c** are representative of three independent experiments.

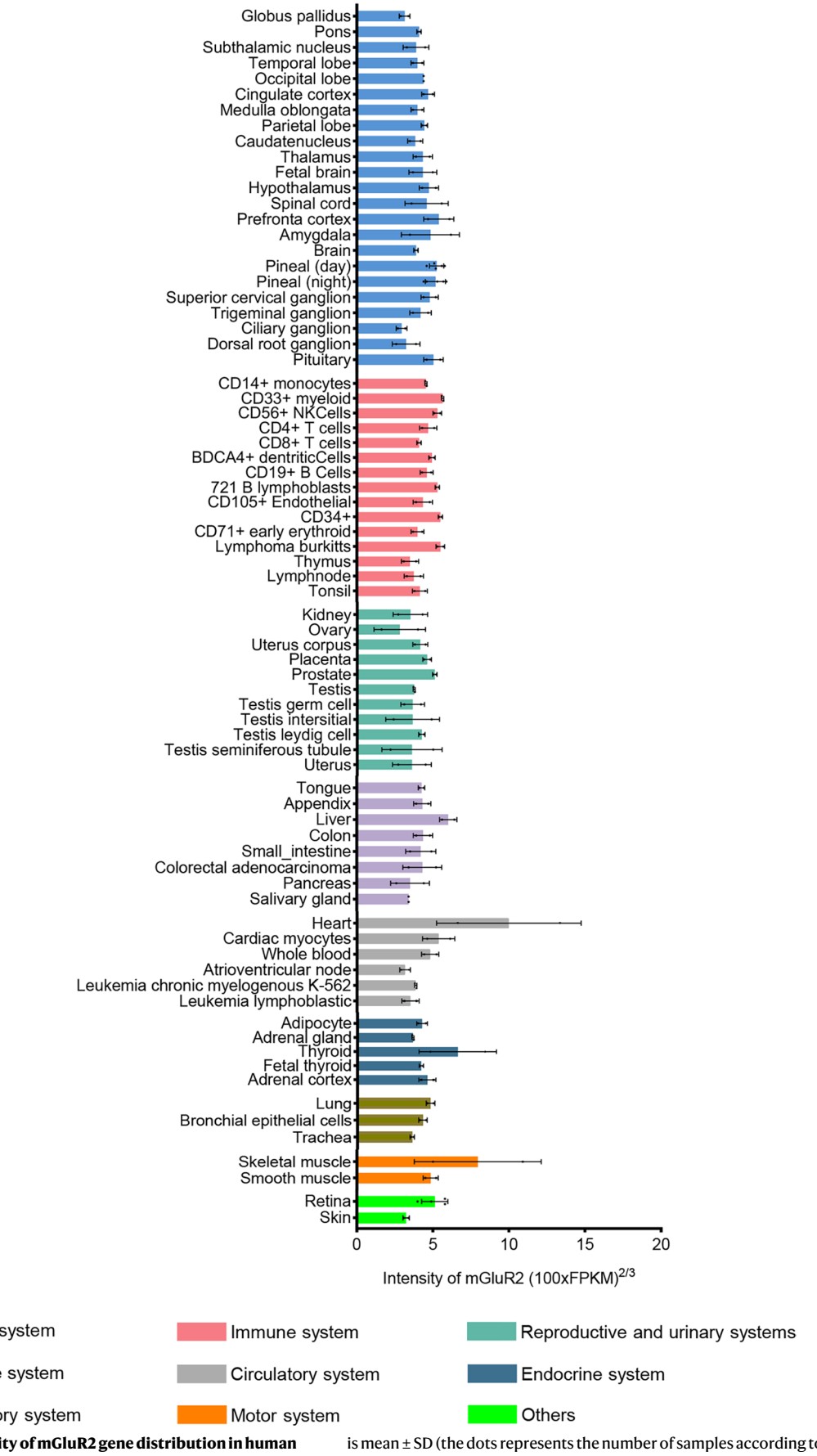

**Extended Data Fig. 3 | The intensity of mGluR2 gene distribution in human tissues.** The data was summarized from BioGPS database (http://biogps.org). Error bar in panel indicates the standard deviation. The data shown in panel is mean ± SD (the dots represents the number of samples according to the database). For pineal and retina, n = 4 biologically independent samples, for the other tissues, n = 2 biologically independent sample.

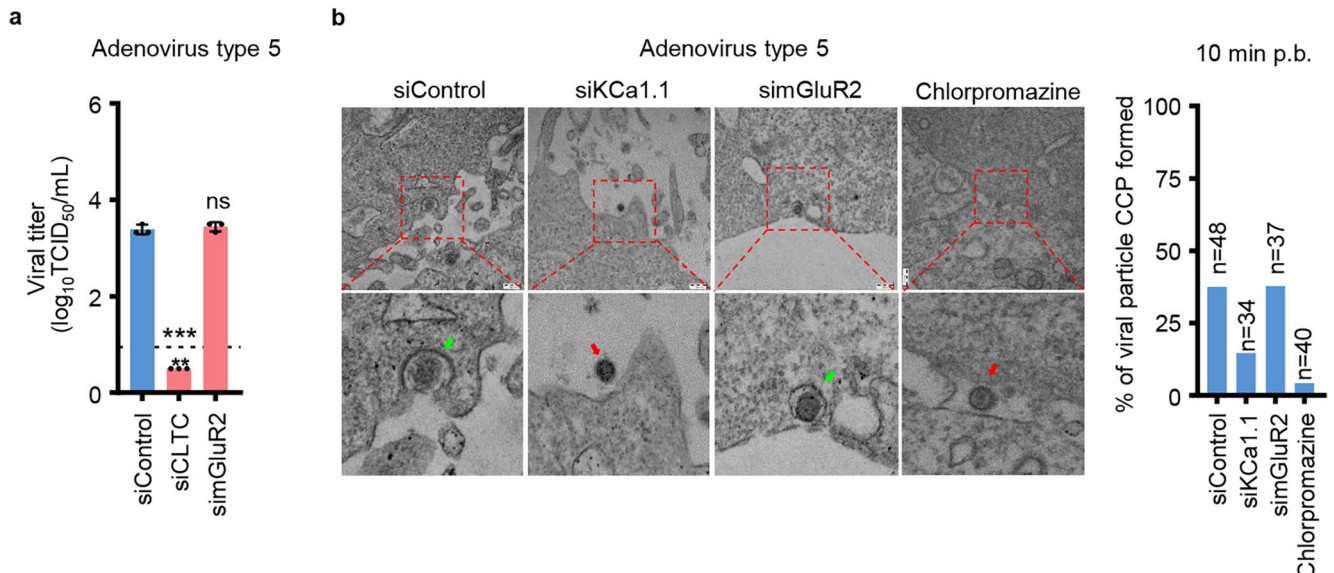

**Extended Data Fig. 4 | mGluR2 did not affect the formation of adenovirus type 5 CCPs. a**, Adenovirus type 5 replication was significantly reduced in CLTC-knockdown A549 cells but not in mGluR2-knockdown A549 cells. Error bar in panels **a** indicates the standard deviation. The data shown in panels **a** is means ± s.d. (n = 3 biologically independent experiments). Statistical analysis was performed by using the unpaired, two-tailed Student's t-test, ns, not significant, ***P < 0.001. Exact P values are available in Source Data. **b**, Formation of adenovirus type 5 CCPs was reduced in KCa1.1-knockdown or chlorpromazine-treated A549 cells but not in mGluR2-knockdown A549 cells. Green arrow, adenovirus type 5 CCPs; red arrow, attached adenovirus type 5. Scale bar, 200 nm. The images in **b** are representative of three independent experiments.

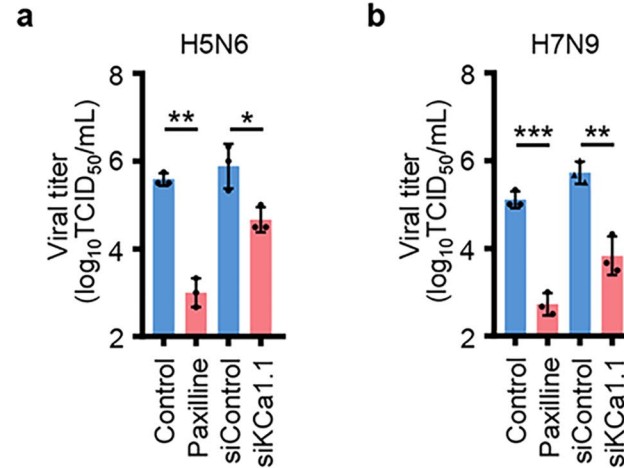

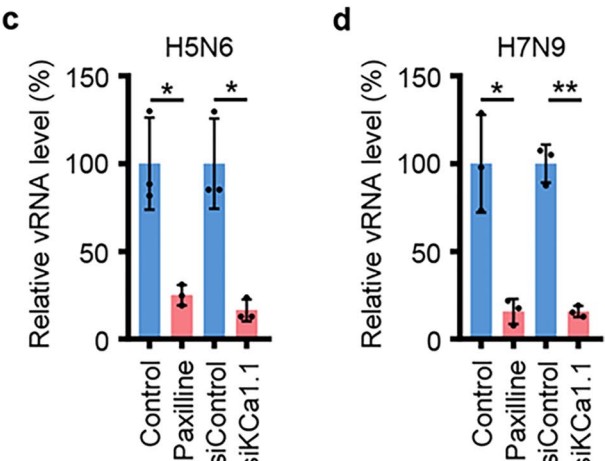

**Extended Data Fig. 5 | Roles of KCa1.1 in H5N6 and H7N9 virus internalization.** **a** and **b**, Replication of H5N6 (**a**) and H7N9 (**b**) viruses in KCa1.1-silenced or paxilline-treated A549 cells. **c** and **d**, Internalization of H5N6 (**c**) and H7N9 (**d**) viruses was significantly reduced in paxilline-treated or KCa1.1-knockdown

A549 cells. Error bar in panels **a**–**d** indicates the standard deviation. The data shown in panels **a**–**d** are means ± s.d. (n = 3 biologically independent experiments). Statistical analysis was performed by using the unpaired, two-tailed Student's t-test, *$P < 0.05$, **$P < 0.01$, ***$P < 0.001$. Exact $P$ values are available in Source Data.

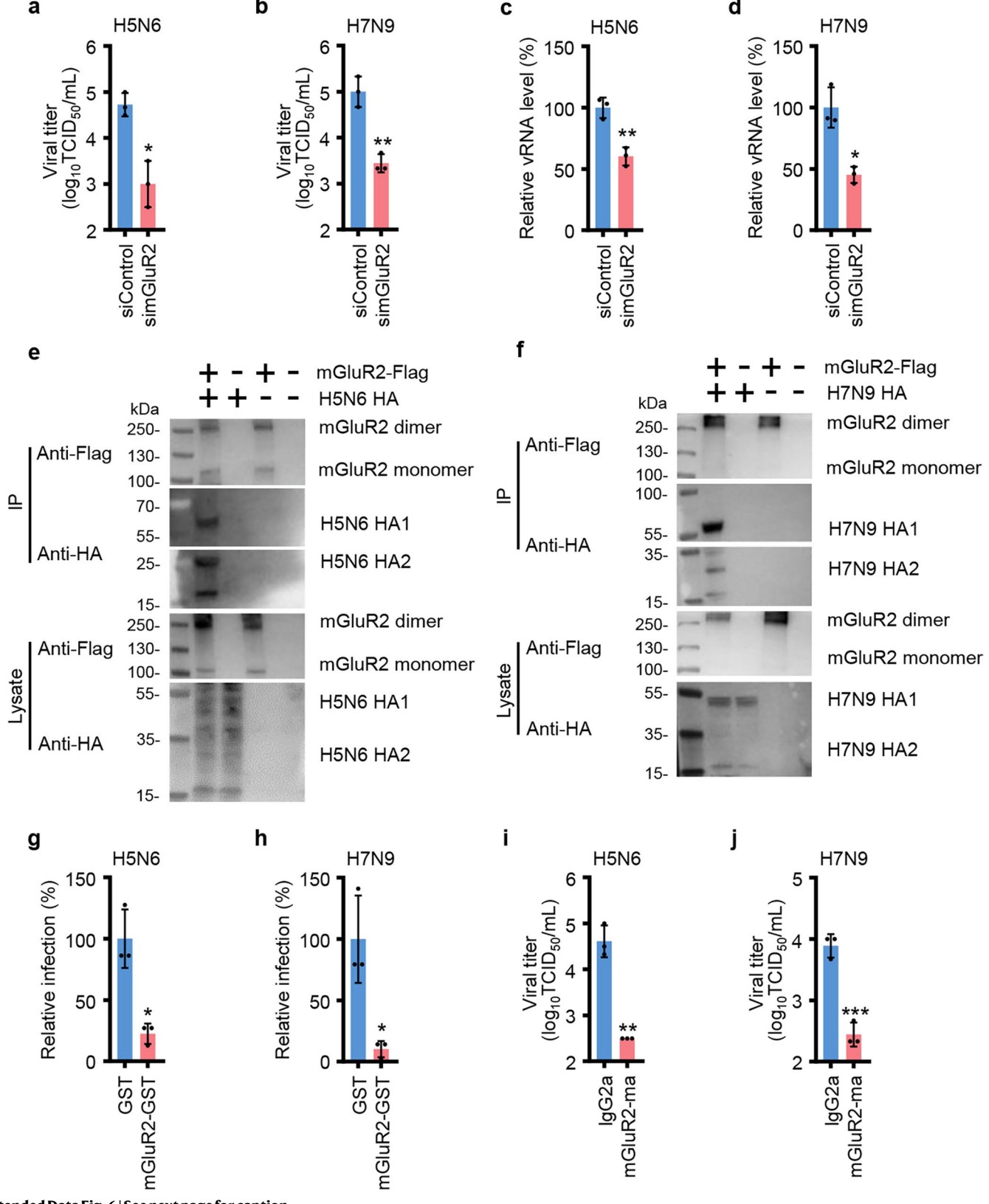

**Extended Data Fig. 6 | See next page for caption.**

**Article** https://doi.org/10.1038/s41564-024-01713-x

**Extended Data Fig. 6 | Roles of mGluR2 in H5N6 and H7N9 virus internalization. a** and **b**, Replication of H5N6 (**a**) and H7N9 (**b**) was significantly impaired in mGluR2-silenced A549 cells. **c** and **d**, Internalization of H5N6 (**c**) and H7N9 (**d**) viruses in mGluR2-knockdown A549 cells was significantly lower than that for control cells. **e**, Interaction of H5 HA and mGluR2 demonstrated by co-immunoprecipitation with the anti-Flag antibody coupled agarose beads. **f**, Interaction of H7 HA and mGluR2 demonstrated by co-immunoprecipitation with the anti-Flag antibody coupled agarose beads. **g**, Soluble protein mGluR2-GST treatment significantly reduced H5N6 virus replication in A549 cells.

**h**, Soluble protein mGluR2-GST treatment significantly reduced H7N9 virus replication in A549 cells. **i**, mGluR2-ma treatment significantly reduced H5N6 virus replication in A549 cells. **j**, mGluR2-ma treatment significantly reduced H7N9 virus replication in A549 cells. The images in **i** and **j** are representative of three independent experiments. Error bar in panels **a**–**d** and **g**–**j** indicates the standard deviation. The data shown in panels **a**–**d** and **g**–**j** are means ± s.d. (n = 3 biologically independent experiments). Statistical analysis was performed by using the unpaired, two-tailed Student's t-test, *$P < 0.05$, **$P < 0.01$, ***$P < 0.001$. Exact $P$ values are available in Source Data.

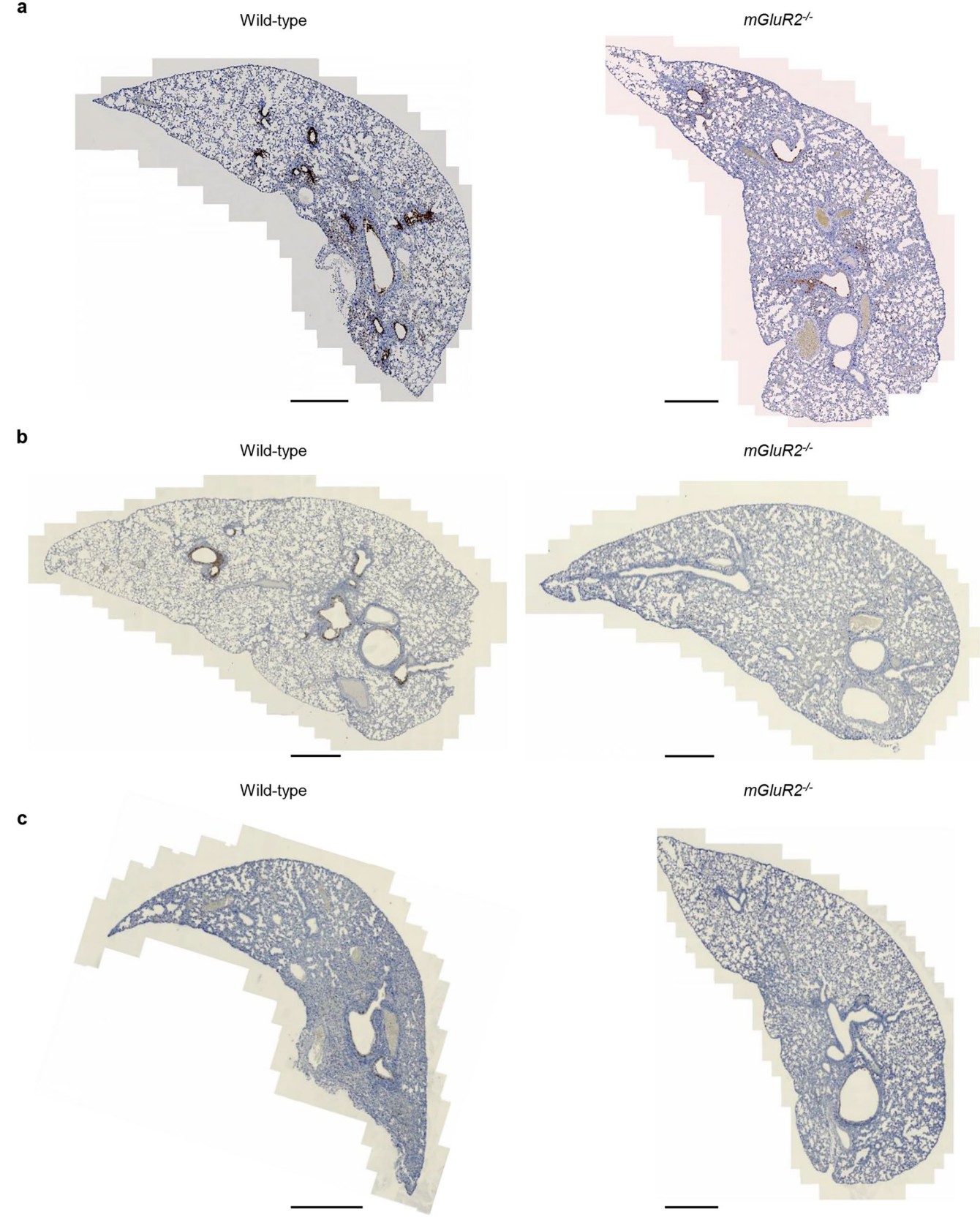

**Extended Data Fig. 7 | Viral antigen detected in the lungs of mice infected by influenza virus.** Viral antigen in the lungs of wild-type and *mGluR2⁻/⁻* mice that were euthanized on day 3 p.i. intranasally infected with 10 MLD$_{50}$ of H1N1 virus (**a**), H5N6 virus (**b**), or H7N9 virus (**c**) were detected by means of immunohistochemical staining. Scale bar, 500 μm. The images in **a**–**c** are representative of three independent experiments.

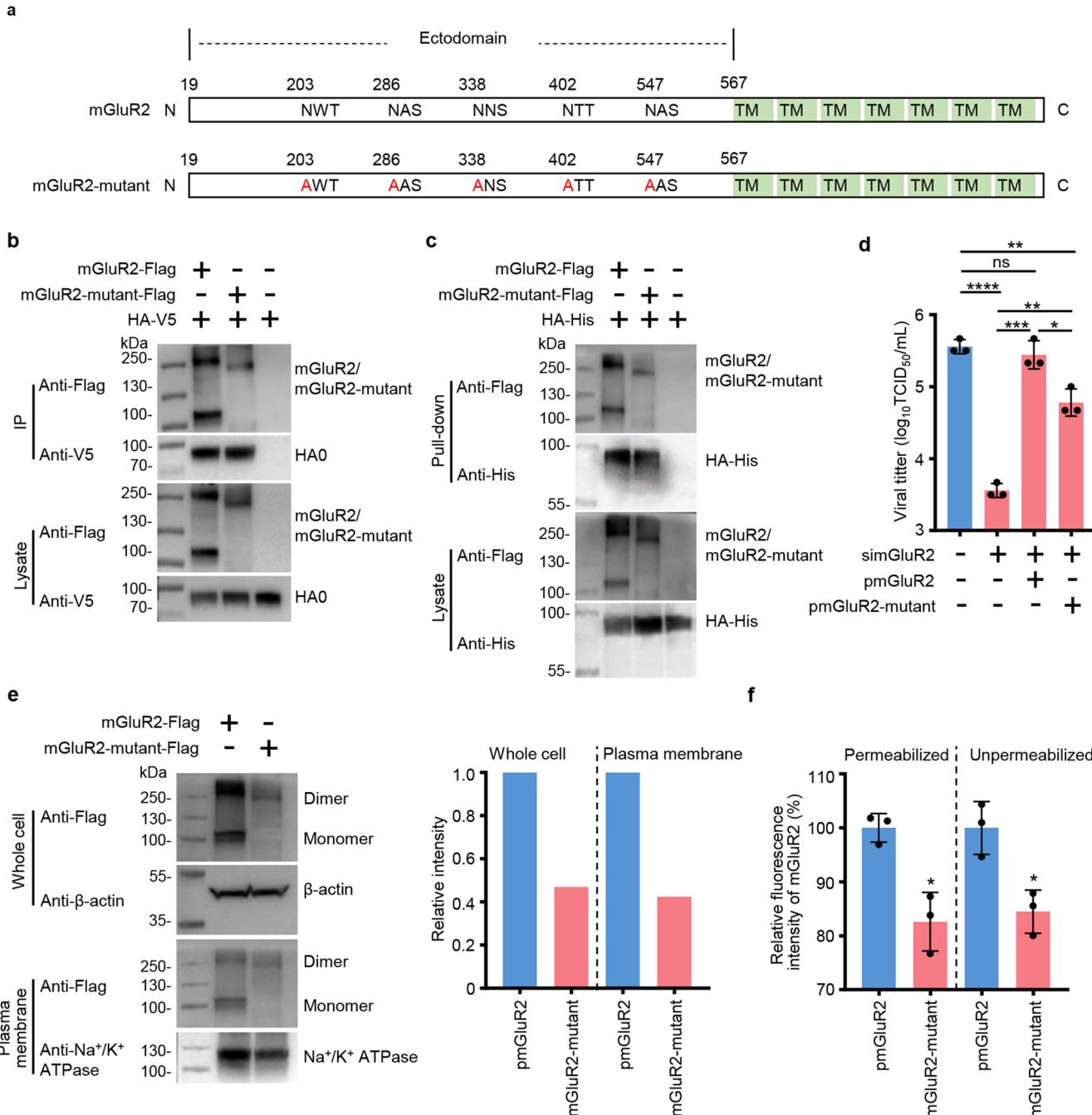

Extended Data Fig. 8 | The N-glycosylation of mGluR2 is dispensable for its interaction with the HA protein. a, Schematic representation of glycosylation mutation sites in the ectodomain of mGluR2. b and c, Interaction between viral HA and mGluR2 or mGluR2-mutant was analyzed by using co-immunoprecipitation (b) and pull-down (c) with the anti-Flag antibody coupled agarose beads. d, Overexpression of pmGluR2-mutant restored influenza virus infection of mGluR2-knockdown A549 cells but not fully. e and f, Abundance of mGluR2 and mGluR2-mutant in whole cells and on the plasma membrane confirmed by western blotting (e) and flow cytometry (f). Control loadings in panel e are run on different gels, the samples derive from the same experiment and that blots were processed in parallel. The images in b, c, and e are representative of three independent experiments. Error bar in panels d and f indicates the standard deviation. The data shown in panels d–f are means ± s.d. (n = 3 biologically independent experiments). Statistical analysis was performed by using the unpaired, two-tailed Student's t-test, ns, not significant, *P < 0.05, **P < 0.01, ***P < 0.001, ****P < 0.0001. Exact P values are available in Source Data.

# Reporting Summary

## Statistics

For all statistical analyses, confirm that the following items are present in the figure legend, table legend, main text, or Methods section.

| n/a | Confirmed | |
|---|---|---|
| ☐ | ☒ | The exact sample size (*n*) for each experimental group/condition, given as a discrete number and unit of measurement |
| ☐ | ☒ | A statement on whether measurements were taken from distinct samples or whether the same sample was measured repeatedly |
| ☐ | ☒ | The statistical test(s) used AND whether they are one- or two-sided *Only common tests should be described solely by name; describe more complex techniques in the Methods section.* |
| ☒ | ☐ | A description of all covariates tested |
| ☒ | ☐ | A description of any assumptions or corrections, such as tests of normality and adjustment for multiple comparisons |
| ☐ | ☒ | A full description of the statistical parameters including central tendency (e.g. means) or other basic estimates (e.g. regression coefficient) AND variation (e.g. standard deviation) or associated estimates of uncertainty (e.g. confidence intervals) |
| ☐ | ☒ | For null hypothesis testing, the test statistic (e.g. $F$, $t$, $r$) with confidence intervals, effect sizes, degrees of freedom and $P$ value noted *Give P values as exact values whenever suitable.* |
| ☒ | ☐ | For Bayesian analysis, information on the choice of priors and Markov chain Monte Carlo settings |
| ☒ | ☐ | For hierarchical and complex designs, identification of the appropriate level for tests and full reporting of outcomes |
| ☒ | ☐ | Estimates of effect sizes (e.g. Cohen's *d*, Pearson's *r*), indicating how they were calculated |

*Our web collection on statistics for biologists contains articles on many of the points above.*

## Software and code

Policy information about availability of computer code

| | |
|---|---|
| Data collection | Images and histochemistry images were acquired by using a confocal laser scanning microscope 980 and 700, STED images were taken by using a Abberior instruments stedycon equipped with an IX83 microscope. Electron microscope images of the process of influenza virus internalization were taken by using Hitachi-7650 transmission electron microscope. |
| Data analysis | GraphPad Prism (version 8.0) was used for data statistical analysis; ZEN software (version 2.3) was used for calculation of fluorescence intensity of images; FlowJo (version 10.0) was used for flow cytometry assay; Image J (version 1.53t) was used to analyze the intensity of protein bands. Stedycon Smart Control software (firmware version 9.0.696-g982382b; FPGA version 13-g2becb89; headboard firmware version 2.11) was used to convert the STED images. |

For manuscripts utilizing custom algorithms or software that are central to the research but not yet described in published literature, software must be made available to editors and reviewers. We strongly encourage code deposition in a community repository (e.g. GitHub). See the Nature Portfolio guidelines for submitting code & software for further information.

## Data

Policy information about availability of data
All manuscripts must include a data availability statement. This statement should provide the following information, where applicable:
- Accession codes, unique identifiers, or web links for publicly available datasets
- A description of any restrictions on data availability
- For clinical datasets or third party data, please ensure that the statement adheres to our policy

> The authors declare that the data supporting the findings of this study are available within the paper and its source data files. The mRNA sequencing data of mGluR2 (Extended Data Fig. 3) in human tissues was from BioGPS database (http://biogps.org/#goto=genereport&id=2912).

## Research involving human participants, their data, or biological material

Policy information about studies with human participants or human data. See also policy information about sex, gender (identity/presentation), and sexual orientation and race, ethnicity and racism.

| | |
|---|---|
| Reporting on sex and gender | not applicable |
| Reporting on race, ethnicity, or other socially relevant groupings | not applicable |
| Population characteristics | not applicable |
| Recruitment | not applicable |
| Ethics oversight | not applicable |

Note that full information on the approval of the study protocol must also be provided in the manuscript.

# Field-specific reporting

Please select the one below that is the best fit for your research. If you are not sure, read the appropriate sections before making your selection.

☒ Life sciences    ☐ Behavioural & social sciences    ☐ Ecological, evolutionary & environmental sciences

For a reference copy of the document with all sections, see [nature.com/documents/nr-reporting-summary-flat.pdf](http://nature.com/documents/nr-reporting-summary-flat.pdf)

# Life sciences study design

All studies must disclose on these points even when the disclosure is negative.

| | |
|---|---|
| Sample size | For all experiments, 3 biological independent experiments for statistical relevance (the unpaired, two-tailed Student's t-test). For mice experiments, 18 mice per group, more than the number of mice usually used for this type of experiment (8 mice). |
| Data exclusions | No data were excluded from the analyses. |
| Replication | 3 independent experiments, consistent data between the experiment, all replication attempts were successful. |
| Randomization | For in vitro study, randomization is not possible because the same cells were treated differently, therefore a random distribution of the samples was not possible. For in vivo study, randomization was not possible because the mice in two groups were genetically different. |
| Blinding | For all experiments, the samples were not blinded but the analysis were blinded (the experimenters that run the experiments and the one that analyzed the data were different, any information of the samples was not showed during the process of analysis). |

# Reporting for specific materials, systems and methods

We require information from authors about some types of materials, experimental systems and methods used in many studies. Here, indicate whether each material, system or method listed is relevant to your study. If you are not sure if a list item applies to your research, read the appropriate section before selecting a response.

## Materials & experimental systems

| n/a | Involved in the study |
|---|---|
| ☐ | ☒ Antibodies |
| ☐ | ☒ Eukaryotic cell lines |
| ☒ | ☐ Palaeontology and archaeology |
| ☐ | ☒ Animals and other organisms |
| ☒ | ☐ Clinical data |
| ☒ | ☐ Dual use research of concern |
| ☒ | ☐ Plants |

## Methods

| n/a | Involved in the study |
|---|---|
| ☒ | ☐ ChIP-seq |
| ☐ | ☒ Flow cytometry |
| ☒ | ☐ MRI-based neuroimaging |

# Antibodies

| | |
|---|---|
| Antibodies used | Rabbit anti-H1-hemagglutinin polyclonal antibody (pAb), Sino Biological, Cat# 11692-T62, microscopy-based assay, 1:300<br>Rabbit anti-Flag-tag pAb, Genscript, Cat# A00170, western blotting assay, 1:1000<br>Rabbit anti-Myc-tag pAb, Genscript, Cat# A00172, western blotting assay, 1:1000; immunoelectron microscopy assay, 1:50<br>Rabbit anti-AQP5 pAb antibody, Boster, Cat# A03085, immunofluorescence histochemistry, 1:100<br>Rabbit anti-6-His pAb, Sigma-Aldrich, Cat# SAB4301134, western blotting assay, 1:1000<br>Goat anti-chicken IgY coupled with HRP, Sigma-Aldrich, Cat# A9046, western blotting assay, 1:2000<br>Horse anti-mouse IgG coupled with HRP, Vectorlabs, Cat# MP-7802-15, microscopy-based assay, immunofluorescence histochemistry, 1:5<br>Goat anti-rabbit IgG coupled with gold (10 nm), Sigma-Aldrich, Cat# G7402, immunoelectron microscopy assay, 1:1000<br>Goat anti-mouse IgG coupled with Alexa Fluor 568, Abcam, Cat# ab175473, microscopy-based assay, 1:300<br>Goat anti-mouse IgG coupled with Alexa Fluor 488, Abcam, Cat# ab150113, microscopy-based assay, 1:300<br>Goat anti-rabbit IgG coupled with Alexa Fluor 488, Invitrogen, Cat# A11034, microscopy-based assay, 1:300<br>Goat anti-rabbit IgG coupled with HRP, Invitrogen, Cat# 31460, western blotting assay, 1:2000<br>Goat anti-mouse Abberior STAR RED, Abberior, Cat# 20831PK-3, STED Microscopy assay, 1:200,<br>Goat anti-mouse Abberior STAR ORANGE, Abberior, Cat# 20831PK-6, STED Microscopy assay, 1:200<br>Goat anti-mouse IgG coupled with Cy3, Beyotime, Cat# A0521, immunofluorescence histochemistry, 1:100<br>Goat anti-rabbit IgG coupled with fluorescein, Vectorlabs, Cat# FI-1000, immunofluorescence histochemistry, 1:200<br>Chicken anti-H5 or H7-hemagglutinin pAb, produced in our lab, western blotting assay, 1:500<br>Rabbit anti-KCa1.1 pAb antibody, Alomone Labs, APC-151, microscopy-based assay, 1:100<br>Rabbit anti-MUC1 pAb antibody, Bioss, Cat# bs-1497R, immunofluorescence histochemistry, 1:100<br>Mouse anti-mGluR2 monoclonal antibody (mAb) [A-1], Santa Cruz Biotechnology, Cat# sc271654, microscopy-based assay,1:200<br>Mouse anti-nucleoprotein mAb antibody [10E9], produced in our lab, immunofluorescence histochemistry, 1:200<br>Mouse IgG2a mAb [HOPC-1], Southern Biotech, Cat# 0103-01, antibody blocking assay, 1:25<br>Rabbit anti-Na+/K+ ATPase mAb [EP1845Y], Abcam, Cat# ab76020, western blotting assay, 1:500<br>Rabbit anti-S100A9 mAb antibody [EPR22332-75], Abcam, Cat# ab242945, immunofluorescence histochemistry, 1:800<br>Rabbit anti-V5-tag mAb [D3H8Q], Cell Signaling, Cat# 13202S, western blotting assay, 1:1000 |
| Validation | All the commercial antibodies have been verified by the manufactures from their websites. Chicken anti-H5 or H7-hemagglutinin pAbs were validated by Yuancheng Zhang (PMID: 38005926) and Xin Yin (PMID:33905456). Mouse anti-nucleoprotein mAb antibody [10E9] was validated by Yuhui Zhao (DOI:10.1016/S2095-3119(21)63840-6). |

# Eukaryotic cell lines

Policy information about cell lines and Sex and Gender in Research

| | |
|---|---|
| Cell line source(s) | HEK293 cells (ATCC, CRL-1573), A549 cells (ATCC, CCL-185), MDCK cells (ATCC, PTA-6500), 293E cells (ATCC, CRL-1573) were purchased from ATCC |
| Authentication | All cells used in this study were verified by ATCC. Cell morphology was monitored at each passage by microscope. We will discard the cells after 15 passages, and recover new cells from frozen stocks. |
| Mycoplasma contamination | All cell lines tested negative for mycoplasma contamination. |
| Commonly misidentified lines<br>(See ICLAC register) | No commonly misidentified lines were used in this study. |

# Animals and other research organisms

Policy information about studies involving animals; ARRIVE guidelines recommended for reporting animal research, and Sex and Gender in Research

| | |
|---|---|
| Laboratory animals | Six-week-old wild-type and mGluR2 gene knockout C57BL/6J mice were used in this study. Mice were housed in ventilated cages (up to 8 mice/cage) with food and water ad libitum. Mice were housed under 12 h light/dark cycles in a controlled environment maintained at 24 degreeCelsius and 40-60% humidity. |

| Wild animals | This study did not involve wild animals. |
|---|---|
| Reporting on sex | Male and female mice were both used in this study. |
| Field-collected samples | This study did not involve samples collected from field. |
| Ethics oversight | Animal studies were carried out in strict accordance with the recommendations in the Guide for the Care and Use of Laboratory Animals of the Ministry of Science and Technology of the People's Republic of China. The protocols were approved by the Committee on the Ethics of Animal Experiments of Harbin Veterinary Research Institute (HVRI) of Chinese Academy of Agricultural Sciences (CAAS) (approval number IACUC-2022-220719-01-GJ). |

Note that full information on the approval of the study protocol must also be provided in the manuscript.

# Plants

| Seed stocks | not applicable |
|---|---|
| Novel plant genotypes | not applicable |
| Authentication | not applicable |

# Flow Cytometry

## Plots

Confirm that:

☒ The axis labels state the marker and fluorochrome used (e.g. CD4-FITC).

☒ The axis scales are clearly visible. Include numbers along axes only for bottom left plot of group (a 'group' is an analysis of identical markers).

☒ All plots are contour plots with outliers or pseudocolor plots.

☒ A numerical value for number of cells or percentage (with statistics) is provided.

## Methodology

| Sample preparation | A549 cells were transfected with pmGluR2 or pmGluR2-mutant for 24 h and collected in a 1.5-mL tube. Then the cells were washed three times with FACS wash buffer (PBS containing 2% FCS) and fixed with 4% PFA at room temperature for 15 min. The cells were then permeabilized with 0.5% Triton X-100 for 30 min (skip this step in the unpermeabilized condition). The permeabilized or unpermeabilized cells were incubated with mouse anti-mGluR2 mAb (1:100) as the primary antibody and goat anti-mouse IgG coupled with Alexa Fluor 488 (1:1000) as the secondary antibody. The cells were analyzed by using a FC-500 flow cytometer (Beckman Coulter). The fluorescence intensity of mGluR2 was analyzed by using Flow Jo software. |
|---|---|
| Instrument | FC500 flow cytometer (Beckman Coulter). |
| Software | FlowJo (version 10.0) |
| Cell population abundance | N/A. Sorting was not performed so not applicable here. |
| Gating strategy | Flow cytometric analyses were gated by FSC, SSC scatters. See diagram of strategy in Source Data. |

☒ Tick this box to confirm that a figure exemplifying the gating strategy is provided in the Supplementary Information.

