## [Peer Review File · Nature Microbiology]

Peer Review Information

Journal: Nature Microbiology

Manuscript Title: Influenza virus uses mGluR2 as an endocytic receptor to enter cells

Corresponding author name(s): Professor Hualan Chen

Reviewer Comments & Decisions:

Decision Letter, initial version:

Message: 19th December 2023

Dear Professor Chen,

Thank you for your patience while your manuscript "Influenza virus uses mGluR2 as an endocytic receptor to enter cells" was under peer-review at Nature Microbiology. It has now been seen by 2 referees, whose expertise and comments you will find at the end of this email. Although they find your work of some potential interest, they have raised a number of concerns that will need to be addressed before we can consider publication of the work in Nature Microbiology.

In particular, please revise the main text to improve clarity and add the data from your point-by-point response to the manuscript, as outlined by reviewer #2. In addition, reviewer #2 raises the unclear physiological relevance of your findings and suggests *in vivo*, *ex vivo* experiments, as well as to look at infection in primary cells. At this stage, we are open to discussing these final requests by reviewer #2 (last point) in the new year, as the journal will be closed over the upcoming holidays.

Should further experimental data allow you to address these criticisms, we would be happy to look at a revised manuscript.

Please include a data availability statement as a separate section after Methods but before references, under the heading "Data Availability". This section should inform readers about

2the availability of the data used to support the conclusions of your study. This information includes accession codes to public repositories (data banks for protein, DNA or RNA sequences, microarray, proteomics data etc...), references to source data published alongside the paper, unique identifiers such as URLs to data repository entries, or data set DOIs, and any other statement about data availability. At a minimum, you should include the following statement: "The data that support the findings of this study are available from the corresponding author upon request", mentioning any restrictions on availability. If DOIs are provided, we also strongly encourage including these in the Reference list (authors, title, publisher (repository name), identifier, year). For more guidance on how to write this section please see:
<http://www.nature.com/authors/policies/data/data-availability-statements-data-citations.pdf>

* If you have not done so already we suggest that you begin to revise your manuscript so that it conforms to our Article format instructions at <http://www.nature.com/nmicrobiol/info/final-submission>. Refer also to any guidelines provided in this letter.

When submitting the revised version of your manuscript, please pay close attention to our [href="https://www.nature.com/nature-portfolio/editorial-policies/image-integrity">Digital Image Integrity Guidelines](https://www.nature.com/nature-portfolio/editorial-policies/image-integrity). and to the following points below:

2Note: This url links to your confidential homepage and associated information about manuscripts you may have submitted or be reviewing for us. If you wish to forward this e-mail to co-authors, please delete this link to your homepage first.

Nature Microbiology is committed to improving transparency in authorship. As part of our efforts in this direction, we are now requesting that all authors identified as 'corresponding author' on published papers create and link their Open Researcher and Contributor Identifier (ORCID) with their account on the Manuscript Tracking System (MTS), prior to acceptance. This applies to primary research papers only. ORCID helps the scientific community achieve unambiguous attribution of all scholarly contributions. You can create and link your ORCID from the home page of the MTS by clicking on 'Modify my Springer Nature account'. For more information please visit please visit www.springernature.com/orcid.

If you wish to submit a suitably revised manuscript we would hope to receive it within 3 months. If you cannot send it within this time, please let us know. We will be happy to consider your revision, even if a similar study has been accepted for publication at Nature Microbiology or published elsewhere (up to a maximum of 3 months).

Yours sincerely,

Reviewer Expertise:

Referee #1: virus entry

Referee #2: virology

Reviewer Comments:

Reviewer #1 (Remarks to the Author):

The authors have revised their manuscript (Ms) entitled "Influenza virus uses mGluR2 as an endocytic receptor to enter cells". They have added new data and clarified aspects of the text. In this manner they have addressed most of my major concerns. The Ms would, however, benefit from addressing the minor comments listed below.

Minor Comments

1. Clarify in legends of all of the HA-mGluR2 interaction figures what antibody is used for the blots in the "pulldown" or IP panels. The reader presumes the labels (e.g., "anti-flag" in those (top) panels are the "precipitating/pulling down" antibody and those indicated for the lysate samples (bottom panels) are antibodies used for blots, but this is not crystal clear.

3This request pertains to Fig. 1d, Fig. 3b, Extended Fig. 4i; Extended Fig. 4j, Extended Fig. 5b, Extended Fig. 5c, and Extended Fig. 5e.

2. Fig. 1d: I think you need to center the 'Anti-Flag' and 'Anti-Myc' texts for the lysate samples.

3. RE: Fig. 1e: The augmentation with co-overexpression of GluR2 and KCa1.1 is modest (but as stated statistically significant). Do 293 cells endogenously express both proteins? State this somewhere in text. If so, that could explain why the enhancement of virus internalization is only modest.

4. L219-222: Rephrase. The vesicles in panels 2-4 are presumed to be clathrin-coated; there is no staining for clathrin, nor (to this reader) is the coat clearly composed of clathrin-based on these images.

5. Fig. 3i legend. Weren't the cells permeabilized to see internalized transferrin? Check statements as to which panels show permeabilized and which show unpermeabilized cells.

6. Lines 372-373: This is an interesting, but not proven, idea; hence it would be more appropriate to say: "suggest" vs. "demonstrate".

7. Extended Data Figure 1; legend. Please state what the positive and negative controls were.

Reviewer #2 (Remarks to the Author):

The manuscript entitled "Influenza virus uses mGluR2 as an endocytic receptor to enter cells" describes the identification of KCa1.1 and mGluR2 as required host factors for influenza A virus (IAV) infection and establishes mGluR2 as internalization receptor. In comparison to a previous version, the manuscript has been greatly improved. The authors have included most controls that were lacking before and thus, the data shown now support the authors' findings much better. The authors have also included information and results from the starting point of the manuscript, namely an RNAi screen for the role of ion channels in IAV infection. A few points remain:

- Fig. 1b: The information that fig. 1b includes the same data as 1a (from the original screen) and is not an independent confirmation should not just be available to me but should be stated in the figure legend.

- The information that expression of mGluR2 has been described for different parts of the respiratory system should not just be shown to the reviewers but be mentioned and referenced in the manuscript.

- The authors have not yet addressed whether IAV infection of cells that are deficient for both, KCa1.1 and mGluR2, is possible and if yes to what level. Similarly, they have not tested which cells get infected in the mGluR2 knockout mice. While they see differences in titers in different organs these could stem from lower overall replication or from altered tropism (at the cellular level). Given that the authors promote their factors as targets for antivirals or other intervention strategies it would be important to know how absence of the receptor affects viral tropism.

Author Rebuttal to Initial comments

Response letter to reviewers' comments (NMICROBIOL-23113101-T)

Reviewer #1:

The authors have revised their manuscript (Ms) entitled “Influenza virus uses mGluR2 as an endocytic receptor to enter cells”. They have added new data and clarified aspects of the text. In this manner they have addressed most of my major concerns. The Ms would, however, benefit from addressing the minor comments listed below.

Minor Comments

1. Clarify in legends of all of the HA-mGluR2 interaction figures what antibody is used for the blots in the “pulldown” or IP panels. The reader presumes the labels (e.g., “anti-flag” in those (top) panels are the “precipitating/pulling down” antibody and those indicated for the lysate samples (bottom panels) are antibodies used for blots, but this is not crystal clear. This request pertains to Fig. 1d, Fig. 3b, Extended Fig. 4i; Extended Fig. 4j, Extended Fig. 5b, Extended Fig. 5c, and Extended Fig. 5e.

Response: Thank you for your suggestion. We have added the information of antibodies used for the blots in the “pulldown” or “IP” panels in the figure legends, as suggested.

2. Fig. 1d: I think you need to center the ‘Anti-Flag’ and ‘Anti-Myc’ texts for the lysate samples.

Response: We assume you are referring to Fig. 2d (now Fig. 2e). We have centered the “Anti-Flag” and “Anti-Myc” texts in Fig. 2e for the lysate samples.

3. RE: Fig. 1e: The augmentation with co-overexpression of GluR2 and KCa1.1 is modest (but as stated statistically significant). Do 293 cells endogenously express both proteins? State this somewhere in text. If so, that could explain why the enhancement of virus internalization is only modest.

Response: Endogenous expression of mGluR2 and KCa1.1 in the cells we used is now mentioned in the revised manuscript as suggested (Page 7, lines 179).

4. L219-222: Rephrase. The vesicles in panels 2-4 are presumed to be clathrin-coated; there is no staining for clathrin, nor (to this reader) is the coat clearly composed of clathrin-based on these images.

Response: We rephrased the text as suggested, and it now reads, “Moreover, by use of immunoelectron microscopy, we detected mGluR2 on the cell membrane of different stages of influenza virus endocytosis, including the cell membrane of influenza virus-containing invagination pits and the cell membrane of matured endocytic vesicles, which are consistent with those of the clathrin-coated pits (CCPs) and clathrin-coated vesicles (CCVs)” (Page 9, lines 231-236).

5. Fig. 3i legend. Weren't the cells permeabilized to see internalized transferrin? Check statements as to which panels show permeabilized and which show unpermeabilized cells.

Response: The experiments presented in Fig. 3i were performed under unpermeabilized conditions. The transferrin internalization assay was performed by using Alexa Fluor 488-labeled transferrin, a purified transferrin directly conjugated to Alexa Fluor 488 fluorophore that can be visualized by confocal laser scanning microscopy under unpermeabilized condition. Thus, after quenching extracellular transferrin with an acidic wash buffer, only the internalized Alexa Fluor 488-labeled transferrin can be observed by confocal laser scanning microscopy under unpermeabilized condition. More details about the experiment have been added in the method section (Page 24, lines 674-675).

6. Lines 372-373: This is an interesting, but not proven, idea; hence it would be more appropriate to say: “suggest” vs. “demonstrate”.

Response: Thank you for your suggestion. We replaced “demonstrates” with “suggests” in the revised manuscript (Page 14, lines 398).

7. Extended Data Figure 1; legend. Please state what the positive and negative controls were.

Response: We have stated the positive and negative controls in figure legends as suggested (Page 38).

Reviewer #2 (Remarks to the Author):

The manuscript entitled “Influenza virus uses mGluR2 as an endocytic receptor to enter cells” describes the identification of KCa1.1 and mGluR2 as required host factors for influenza A virus (IAV) infection and establishes mGluR2 as internalization receptor.

In comparison to a previous version, the manuscript has been greatly improved. The authors have included most controls that were lacking before and thus, the data shown now support the authors’ findings much better. The authors have also included information and results from the starting point of the manuscript, namely an RNAi screen for the role of ion channels in IAV infection. A few points remain:

- Fig. 1b: The information that fig. 1b includes the same data as 1a (from the original screen) and is not an independent confirmation should not just be available to me but should be stated in the figure legend.

Response: Thank you for your suggestion. We have revised the figure legend of Fig. 1b by adding “The data were summarized from the screen by combining the results from different siRNAs presented in Fig. 1a.” (Page 30).

- The information that expression of mGluR2 has been described for different parts of the respiratory system should not just be shown to the reviewers but be mentioned and referenced in the manuscript.

*Response: Thank you for your suggestion. The information of distribution of mGluR2 in human tissues is now mentioned in the revised manuscript, and we also provided an **Extended Data Fig. 3** to summarize the data from the public database (Page 8, lines 219-220).*

- The authors have not yet addressed whether IAV infection of cells that are deficient for both, KCa1.1 and mGluR2, is possible and if yes to what level.

Response: If KCa1.1 and mGluR2 independently mediate influenza virus infection via independent mechanisms, simultaneous blocking of KCa1.1 and mGluR2 will have an additive effect on the reduction of influenza virus replication. Our study indicated that the KCa1.1 and mGluR2 mediate CME of influenza virus through an interdependent process. Therefore, we speculate that simultaneous blocking of KCa1.1 and mGluR2 will have no additive effect on the reduction of influenza virus replication. To answer your question, we performed a study to investigate the viral replication in cells with different treatments, and found that viral titers in the paxilline-treated mGluR2-silenced A549 cells were comparable to that of the paxilline-treated cells. We now provide this result in the revised manuscript (Page 6 and 7, lines 165-174; Fig. 2d).

Similarly, they have not tested which cells get infected in the mGluR2 knockout mice. While they see differences in titers in different organs these could stem from lower overall replication or from altered tropism (at the cellular level). Given that the authors promote their factors as targets for antivirals or other intervention strategies it would be important to know how absence of the receptor affects viral tropism.

Response: Our study demonstrated that mGluR2-mediated CME is the most important way but not the only way for influenza virus to enter cells, therefore, mGluR2-knockout may not be able to alter the viral tropism. As you suggested, we performed immunohistologic study and dual-staining analysis with the lung samples of wild-type and mGluR2^{-/-} mice that were infected with different influenza viruses, and found that the viral antigen positive cells in the lungs of mGluR2^{-/-} mice were notably less than that in the wild-type mice, but viral antigen could be detected in type II alveolar epithelial cells, type I epithelial cells, and macrophages in the lungs of both wild-type and mGluR2^{-/-} mice. These results suggest that mGluR2 knockout reduced the overall replication level of influenza virus, but did not affect the viral cell tropism. We now provide these results in the revised manuscript (Page 13, lines 355-365; Fig. 6; Extended Data Fig. 6).

Decision Letter, first revision:

Message: Our ref: NMICROBIOL-23113101A

3rd April 2024

Dear Dr. Chen,

Thank you for your patience as we've prepared the guidelines for final submission of your Nature Microbiology manuscript, "Influenza virus uses mGluR2 as an endocytic receptor to enter cells" (NMICROBIOL-23113101A). Please carefully follow the step-by-step instructions provided in the attached file, and add a response in each row of the table to indicate the changes that you have made. Please also check and comment on any additional marked-up edits we have proposed within the text. Ensuring that each point is addressed will help to ensure that your revised manuscript can be swiftly handed over to our production team.

In recognition of the time and expertise our reviewers provide to Nature Microbiology's editorial process, we would like to formally acknowledge their contribution to the external peer review of your manuscript entitled "Influenza virus uses mGluR2 as an endocytic receptor to enter cells". For those reviewers who give their assent, we will be publishing their names alongside the published article.

Nature Microbiology offers a Transparent Peer Review option for new original research manuscripts submitted after December 1st, 2019. As part of this initiative, we encourage our authors to support increased transparency into the peer review process by agreeing to have the reviewer comments, author rebuttal letters, and editorial decision letters published as a Supplementary item. When you submit your final files please clearly state in your cover letter whether or not you would like to participate in this initiative. Please note that failure to state your preference will result in delays in accepting your manuscript for publication.

9Cover suggestions

COVER ARTWORK: We welcome submissions of artwork for consideration for our cover. For more information, please see our guide for cover artwork.

Nature Microbiology has now transitioned to a unified Rights Collection system which will allow our Author Services team to quickly and easily collect the rights and permissions required to publish your work. Approximately 10 days after your paper is formally accepted, you will receive an email in providing you with a link to complete the grant of rights. If your paper is eligible for Open Access, our Author Services team will also be in touch regarding any additional information that may be required to arrange payment for your article.

Please note that *Nature Microbiology* is a Transformative Journal (TJ). Authors may publish their research with us through the traditional subscription access route or make their paper immediately open access through payment of an article-processing charge (APC). Authors will not be required to make a final decision about access to their article until it has been accepted. Find out more about Transformative Journals

Nature Microbiology

Reviewer #2:

Remarks to the Author:

The authors have addressed the remaining points comprehensively and thereby further improved the manuscript.

10Final Decision Letter:

Message: 24th April 2024

Dear Hualan,

I am delighted to accept your Article "Influenza virus uses mGluR2 as an endocytic receptor to enter cells" for publication in Nature Microbiology. Thank you for having chosen to submit your work to us and many congratulations.

11Please note that *Nature Microbiology* is a Transformative Journal (TJ). Authors may publish their research with us through the traditional subscription access route or make their paper immediately open access through payment of an article-processing charge (APC). Authors will not be required to make a final decision about access to their article until it has been accepted. Find out more about Transformative Journals

Congrats again to you and your co-authors! We are looking forward to seeing your paper

published.

With kind regards,